# Protein-lipid interaction at low pH induces oligomerization of the MakA cytotoxin from *Vibrio cholerae*

**Aftab Nadeem**[1,2,3*†], **Alexandra Berg**[1,2,3,4], **Hudson Pace**[2,5,6], **Athar Alam**[2,3,5], **Eric Toh**[1,2,3], **Jörgen Ådén**[2,7], **Nikola Zlatkov**[1,2,3], **Si Lhyam Myint**[1,2,3], **Karina Persson**[2,7], **Gerhard Gröbner**[2,7], **Anders Sjöstedt**[2,3,5], **Marta Bally**[2,5,6], **Jonas Barandun**[1,2,3], **Bernt Eric Uhlin**[1,2,3*†], **Sun Nyunt Wai**[1,2,3*†]

[1]Department of Molecular Biology, Umeå University, Umeå, Sweden; [2]Umeå Centre for Microbial Research (UCMR), Umeå University, Umeå, Sweden; [3]The Laboratory for Molecular Infection Medicine Sweden (MIMS), Umeå University, Umeå, Sweden; [4]Science for Life Laboratory (SciLifeLab), Department of Molecular Biology, Umeå University, Umeå, Sweden; [5]Department of Clinical Microbiology, Umeå University, Umeå, Sweden; [6]Wallenberg Centre for Molecular Medicine, Umeå University, Umeå, Sweden; [7]Department of Chemistry, Umeå University, Umeå, Sweden

**\*For correspondence:**
aftab.nadeem@umu.se (AN);
bernt.eric.uhlin@umu.se (BEU);
sun.nyunt.wai@umu.se (SNW)

†These authors contributed equally to this work

**Abstract** The α-pore-forming toxins (α-PFTs) from pathogenic bacteria damage host cell membranes by pore formation. We demonstrate a remarkable, hitherto unknown mechanism by an α-PFT protein from *Vibrio cholerae*. As part of the MakA/B/E tripartite toxin, MakA is involved in membrane pore formation similar to other α-PFTs. In contrast, MakA in isolation induces tube-like structures in acidic endosomal compartments of epithelial cells in vitro. The present study unravels the dynamics of tubular growth, which occurs in a pH-, lipid-, and concentration-dependent manner. Within acidified organelle lumens or when incubated with cells in acidic media, MakA forms oligomers and remodels membranes into high-curvature tubes leading to loss of membrane integrity. A 3.7 Å cryo-electron microscopy structure of MakA filaments reveals a unique protein-lipid superstructure. MakA forms a pinecone-like spiral with a central cavity and a thin annular lipid bilayer embedded between the MakA transmembrane helices in its active α-PFT conformation. Our study provides insights into a novel tubulation mechanism of an α-PFT protein and a new mode of action by a secreted bacterial toxin.

## Editor's evaluation

This paper provides some remarkable insights about the pore-forming toxin MakA, from *V. cholerae*, showing how it alone can bind to membranes and form regular tubes. Given the general interest in pore-forming toxins in terms of both understanding bacterial pathogenesis and designing new therapeutics, the paper should be of broad interest.

## Introduction

Several bacterial protein toxins are known to interact directly with target cell membranes by binding to specific receptors, lipids, or proteins on host cell membranes (*Geny and Popoff, 2006*; *Lemichez and Barbieri, 2013*). Both extracellular and intracellular bacterial pathogens produce and secrete host membrane-attacking pore-forming toxins (PFTs) to counteract host defenses, to promote colonization and spread, and to kill other bacteria (*Los et al., 2013*; *Verma et al., 2021*). PFTs can be categorized

into two main groups, α-PFTs and β-PFTs, based on whether the secondary structure of the membrane-penetrating domain contains α-helices or β-barrels, respectively (*Los et al., 2013*). Generally, PFTs are secreted from bacteria in a monomeric, soluble form. Upon recognition and binding to the target cell membrane, the toxins undergo a conformational change, interact with the membrane, dimerize, and oligomerize, leading to the formation of membrane pores (*Bischofberger et al., 2012*).

*Vibrio cholerae*, a Gram-negative bacterium, is the causal organism of the diarrheal disease cholera (*Clemens et al., 2017*). Cholera toxin (CT) and toxin co-regulated pilus (TCP) are the main virulence factors of *V. cholerae* that cause disease in mammalian hosts (*Kaper et al., 1994*; *Taylor et al., 1987*). Most environmental *V. cholerae* isolates do not produce CT and TCP (*Rajpara et al., 2013*). Nevertheless, these bacteria are considered pathogenic since they have been associated with secretory diarrhea and may cause wound infections and sepsis (*Schwartz et al., 2019*). *V. cholerae* strains lacking the CT-encoding genes often contain a set of genes coding for other secreted virulence factors, including hemolysin, hemagglutinin protease, RTX toxin, and multiple lipases that together may play a role in pathogenesis (*Schwartz et al., 2019*).

Using *Caenorhabditis elegans* and *Danio rerio* (zebrafish) as host models for bacterial predatory interactions and infection in aqueous environments, respectively, we obtained evidence for a new *V. cholerae* cytotoxin denoted MakA (motility-associated killing factor A) (*Dongre et al., 2018*). The *V. cholerae* gene *makA* (*vca0883*) is localized in a gene cluster together with *makC* (*vca0881*), *makD* (*vca0880*), *makB* (*vca0882*), and *makE* (*vca0884*). This gene cluster has been found in different *V. cholerae* strains, including CT-negative isolates (*Dongre et al., 2018*; *Nadeem et al., 2021c*; *Tsou and Zhu, 2010*). The crystal structure of MakA revealed that it belongs to the ClyA α-PFT family (*Dongre et al., 2018*), named after the potent, one-component, PFT ClyA which is expressed from a monocistronic operon in *Escherichia coli* (*Oscarsson et al., 1999*; *Oscarsson et al., 1996*). Our recent studies of the proteins encoded by the *mak* genes in *V. cholerae* demonstrated that MakA can form a tripartite cytolytic complex with MakB and MakE, whereas neither of the three proteins displays cytolytic activity on their own (*Nadeem et al., 2021a*). Other family members of α-PFT are found among the bipartite toxins from *Yersinia enterocolitica* (YaxAB) and *Xenorhabdus nematophila* (XaxAB) (*Brauning et al., 2018*; *Schubert et al., 2018*), as well as among the tripartite toxins from *Bacillus cereus* (NheABC and HblL$_1$L$_2$B) (*Sastalla et al., 2013*), *Aeromonas hydrophila* (AhlABC) (*Wilson et al., 2019*), and *Serratia marcescens* (SmhABC) (*Churchill-Angus et al., 2021*). The bipartite and tripartite PFTs require the combined action of all protein partners to induce pore formation in the target membranes, and there is evidence that protein interaction occurs in a specific order for maximum cytolytic activity (*Churchill-Angus et al., 2021*; *Nadeem et al., 2021c*; *Wilson et al., 2019*). We recently demonstrated that an equimolar mixture of the MakA, MakB, and MakE proteins efficiently could assemble into a pore-forming complex in mammalian membranes (*Nadeem et al., 2021c*). However, there are still questions about (i) how many molecules of each subunit protein are required to form a pore, (ii) how they exactly interact with each other, (iii) how structural conformational changes occur, and (iv) how protein moieties are involved in the interaction with the host membrane. In addition, it remains possible that some of the subunit proteins can be separately released from the bacteria and thereby exhibit biological effects on their own. Secretion of the MakA/B/E proteins from *V. cholerae* was shown to be facilitated via the bacterial flagellum (*Dongre et al., 2018*; *Nadeem et al., 2021c*). However, about 10% of MakA was secreted from *V. cholerae* lacking the flagellum suggesting an alternative route of secretion, in contrast to MakB and MakE, which displayed a more definitive flagellum-dependent secretion (*Dongre et al., 2018*; *Nadeem et al., 2021c*).

Our earlier studies with purified MakA protein and cultured mammalian cells showed that the protein binds to the target cell membrane and, upon internalization, may accumulate in the endolysosomal membrane, causing lysosomal dysfunction, modulation of autophagy, and apoptotic cell death (*Corkery et al., 2021*; *Nadeem et al., 2021a*; *Nadeem et al., 2021b*). Cell membrane dynamics are crucial for various biological processes in all kingdoms of life. In eukaryotes, the lipid bilayer and the number of membrane-associated proteins play a crucial role in the regulation of endocytosis, exocytosis, cell motility, cytokinesis, and maintenance of the organelle function in living cells (*Tanaka-Takiguchi et al., 2013*). The major proteins that play a key role in the regulation of endocytosis and intracellular trafficking include F-BAR, ESCRT-III, and dynamin proteins. In vitro, these proteins may associate with lipid membranes and generate spiral filaments, causing the lipid membranes to distort and form tubular assemblies (*McCullough et al., 2018*). Recently, a bacterial protein designated BdpA

(BAR domain-like protein A) with BAR domain-like activity was identified in the *Shewanella oneidensis* MR-1 strain. BdpA forms redox-active membrane vesicles and micrometer-scale outer membrane extensions (*Phillips et al., 2021*). In addition to BAR domain proteins, dynamin was previously shown to be associated with the lipid membrane in vitro and form a helical tube-like assembly (*Sweitzer and Hinshaw, 1998*). Bacterial dynamin-like proteins (BDLPs), like eukaryotic dynamin proteins, are well known for their capacity to induce tube assembly in the presence of lipid membranes (*Low and Löwe, 2006*). Notably, there are only a few known bacterial toxins that produce invaginations in lipid membranes and eukaryotic cells, which can be exploited to transport these toxins intracellularly (*Berg Klenow et al., 2020*).

The present study aimed to further characterize the mechanism(s) behind MakA-induced lysosomal membrane tubulation as well as the pH-dependent molecular interaction(s) between MakA and host cell membranes. We found that under low pH conditions, MakA bound to purified lysosomes and to liposomes prepared from total epithelial cell lipid extracts (ECLE). The insertion of MakA into lysosomes and ECLE liposomes resulted in the formation of tubular assemblies. Cryo-electron microscopy (cryo-EM) analysis of these assemblies revealed an unusual helical structure formed by MakA and lipid spirals. The observed structure revealed that MakA monomers adopted conformational arrangements typical of active membrane-bound α-PFTs while assembling into an atypical polymeric superstructure. Large structural rearrangements, presumably induced by the lowered pH, were necessary for the transition from the inactive soluble form to the extended active toxin form. Interaction of these MakA structures with cell membranes caused cell death in the in vitro setting. MakA is the first *V. cholerae* protein that engages target membranes to form nanotubes by polymerizing as a helical structure together with a lipid spiral.

## Results

### pH-dependent formation of tubular structures in lysosomes and on cell membranes by MakA

In recent in vitro studies with epithelial cells, we found that internalized MakA protein accumulated in the acidic endolysosomal compartment where it caused formation of tube-like structures and induced lysosomal permeability (*Nadeem et al., 2021b*). Our findings prompted us to examine how the observed lysosomal tubulation and lysosomal dysfunction might be caused by MakA. Upon treatment of Caco-2 cells with MakA (250 nM, 18 hr), we observed co-localization of the Alexa568-labeled MakA (Alexa568-MakA) with GFP-LAMP1 or lysotracker in tubular structures (*Figure 1A* and *Figure 1—figure supplement 1A*). The tubulation in the lysosomes was further confirmed by transmission electron microscopy (TEM) of lysosomes isolated from MakA-treated HCT8 cells (*Figure 1B*). Western blot analysis of lysosomes isolated from MakA-treated (250nM, 18 hr) HCT8 and Caco-2 cells revealed, in addition to the MakA monomer, the formation of dimeric, tetrameric, and oligomeric MakA complexes (*Figure 1—figure supplement 1B*). To investigate if MakA can also induce tubulation of lysosomes outside the intracellular environment, we purified lysosomes from untreated HCT8 cells and exposed them to native MakA or to Alexa568-MakA. Both confocal microscopy and TEM analysis revealed aggregation and tubulation of lysosomes at pH 5.0 (*Figure 1C and D*). In addition, a majority of the lysosomes showed well-organized tubulation when exposed to MakA at pH 6.5 (*Figure 1D*). In contrast, we did not observe any MakA-induced tubular structures in lysosomes at pH 7.0 (*Figure 1D*). Western blot analysis confirmed pH-dependent binding of MakA to lysosomes (*Figure 1E*). Alexa568-MakA was subsequently shown to bind epithelial cells in a pH-dependent manner (*Figure 1F and G*). To determine the kinetics of MakA binding to the target cells, HCT8 cells were exposed to Alexa568-MakA at pH 5.0, and live-cell imaging was performed using spinning disc confocal microscopic analysis. Consistent with our earlier findings (*Nadeem et al., 2021a*), we observed accumulation of Alexa568-MakA on the plasma membrane, including filipodia-rich tubular structures, in a time-dependent manner (*Figure 1H* and *Figure 1—figure supplement 1C, D*). The time scale of MakA binding to individual HCT8 cells ranged from ~40 min to 4 hr after Alexa568-MakA treatment (*Figure 1F–H* and *Figure 1—figure supplement 1C, D*). Ultimately, Alexa568-MakA was detected on the plasma membrane of the entire cell population, with most cells positive for tubular structures protruding out from the plasma membrane (*Figure 1F–H*). Taken together, these results

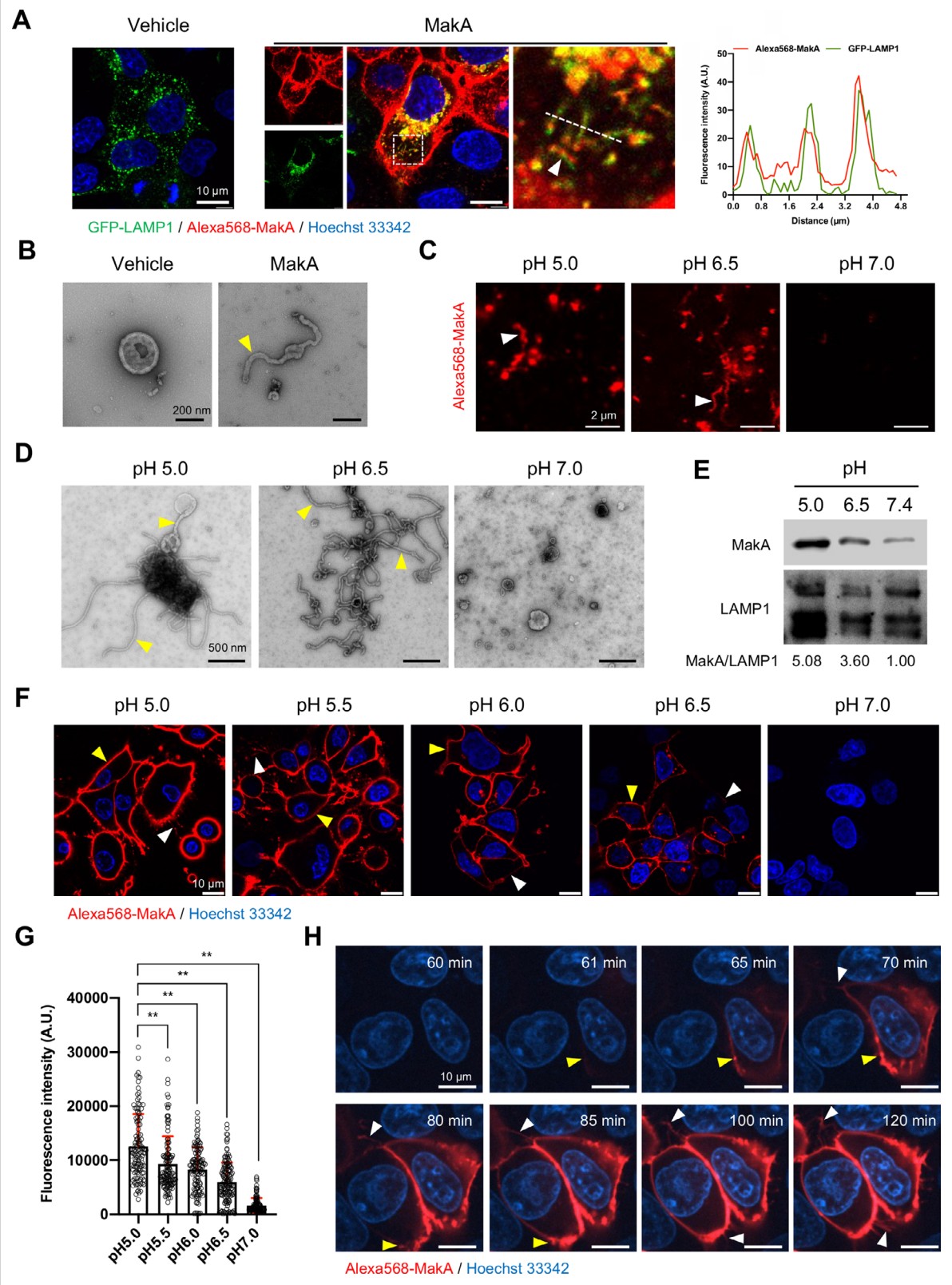

**Figure 1.** pH-dependent tubulation of lysosomes and binding to epithelial cells by MakA. (**A**) Caco-2 cells transfected with GFP-LAMP1 were treated with vehicle or Alexa568-MakA (250 nM, 18 hr). Nuclei were counterstained with Hoechst 33342. The line graph to the right indicates the accumulation of Alexa568-MakA in GFP-LAMP1-positive tubular lysosomes. Pearson correlation coefficient was used to calculate Alexa568-MakA (red) co-localization with GFP-LAMP1 (green) along with the tubular structures. Scale bars, 10 µm. (**B**) Representative electron micrographs of lysosomes purified from

*Figure 1 continued on next page*

*Figure 1 continued*

vehicle and MakA- (250 nM, 24 hr) treated HCT8 cells. Scale bars, 200 nm. The yellow arrowhead indicates a tubular structure found with lysosomes from MakA-treated cells. (**C**) The pH-dependent binding of Alexa568-MakA to purified lysosomes isolated from HCT8 cells: White arrowheads point to the tubular structures observed with MakA-treated purified lysosomes. Images shown for specimens from different pH conditions were acquired using the same settings of the microscope. Scale bars, 2 µm. (**D**) Representative electron micrographs of purified lysosomes treated with MakA (1 µM) under different pH conditions. Yellow arrowheads indicate tubular structures that appear at low pH. Scale bars, 500 nm. (**E**) Western blot analysis of samples from lysosome pull-down assay treated with MakA (250 nM, 60 min) under different pH conditions. Lysosome-bound MakA was detected with anti-MakA antiserum. Immunodetection of LAMP1 was used as a reference and the MakA/LAMP1 ratio was determined for the quantification of relative MakA amounts. (**F**) HCT8 cells were exposed to Alexa568-MakA (500 nM, 4 hr) under different pH conditions and visualized live under confocal microscopy. Nuclei were counterstained with Hoechst 33342 (blue). Yellow arrowheads indicate cell membrane association, while white arrowheads indicate MakA-positive tubular structures. The different images were acquired using the same microscope settings. Scale bars, 10 µm. (**G**) The histogram indicates quantification of cell-bound Alexa568-MakA (n = 100 cells) as shown in (**F**). Data from two independent experiments are presented as mean ± s.e.m.; one-way analysis of variance (ANOVA) with Dunnett's multiple comparisons test. **p ≤ 0.01. (**H**) Still images of HCT8 cells exposed to Alexa568-MakA (500 nM) at pH 5.0. Yellow arrowheads indicate the initial binding site of MakA and white arrowheads indicate the appearance of MakA-positive tubular structures in a time-dependent manner. Nuclei were counterstained with Hoechst 33342. Scale bars, 10 µm.

The online version of this article includes the following source data and figure supplement(s) for figure 1:

**Source data 1.** Data used to calculate Pearson correlation coefficient.

**Source data 2.** Original Western blots for *Figure 1E*.

**Source data 3.** Quantification of cell-bound Alexa568-MakA.

**Figure supplement 1.** MakA binding to the epithelial cell membrane in filipodia rich structures.

**Figure supplement 1—source data 1.** Original Western blots for MakA.

suggest that MakA can cause tubulation of both endolysosomal membranes and plasma membranes in a pH-dependent manner.

## pH-dependent epithelial cell toxicity and formation of tubular structures on erythrocytes by MakA

To further assess the effect of pH on the binding of MakA to HCT8 epithelial cells, we performed Western blot analysis using anti-MakA antiserum (*Figure 2A*). We observed that MakA bound to the cells in a pH-dependent manner and that there seemed to be a pH-dependent formation of stable MakA oligomers bound to the epithelial cells. To determine if MakA binding and oligomerization at the target cell membrane correlated with cytotoxic effect, HCT8 cells were exposed to MakA, and cell toxicity was quantified by a propidium iodide uptake assay using flow cytometry and confocal microscopy (*Figure 2B* and *Figure 2—figure supplement 1A*). In addition to causing pH-dependent toxicity of HCT8 cells, MakA was similarly toxic to other colon cancer cells, Caco-2 and HCT116 cells, as revealed by an increase in the number of trypan blue-positive cells in response to an increasing concentration of MakA in low pH conditions (*Figure 2—figure supplement 1B*).

Erythrocytes are widely used as a cell model to investigate the cytolytic activity of the toxins that belong to the ClyA PFTs family (*Beecher and Wong, 1997*; *Oscarsson et al., 1996*; *Sastalla et al., 2013*; *Schubert et al., 2018*; *Wilson et al., 2019*). To test if pH-dependent binding and oligomerization of MakA may cause hemolysis of erythrocytes, human erythrocytes were exposed to increasing concentrations of MakA at different pH conditions for either 90 min or 5 hr (*Figure 2C*). When erythrocytes were exposed to MakA at pH 5.0, hemolysis was observed in a concentration-dependent manner within 90 min (*Figure 2C*). In contrast, MakA failed to induce hemolysis of erythrocytes at pH 6.5 or pH 7.4 during the 90 min treatment. A detectable, but low level of hemolysis was observed after 5 hr with erythrocytes exposed to MakA at pH 6.5 (*Figure 2C*). With confocal microscopy, we observed pH-dependent binding of Alexa568-MakA to erythrocytes. A majority of erythrocytes at pH 5.0 and 6.5 were covered by Alexa568-MakA, whereas there was virtually no MakA binding observed at pH 7.4 (*Figure 2D* and *Figure 2—figure supplement 1C*). Notably, we detected the presence of tubular structures in association with MakA at the red blood cell surface, as shown by a maximum 3D projection of the z-stack images of erythrocytes (*Figure 2E*). The presence of MakA-induced tubular structures on the surface of erythrocytes was further observed by TEM and scanning electron microscopy (SEM) (*Figure 2F and G* and *Figure 2—figure supplement 1D*). Together, these results suggest that MakA could accumulate in a pH-dependent manner at the surface of both epithelial cells and erythrocytes, thereby inducing formation of tubular structures that ultimately lead to cell lysis.

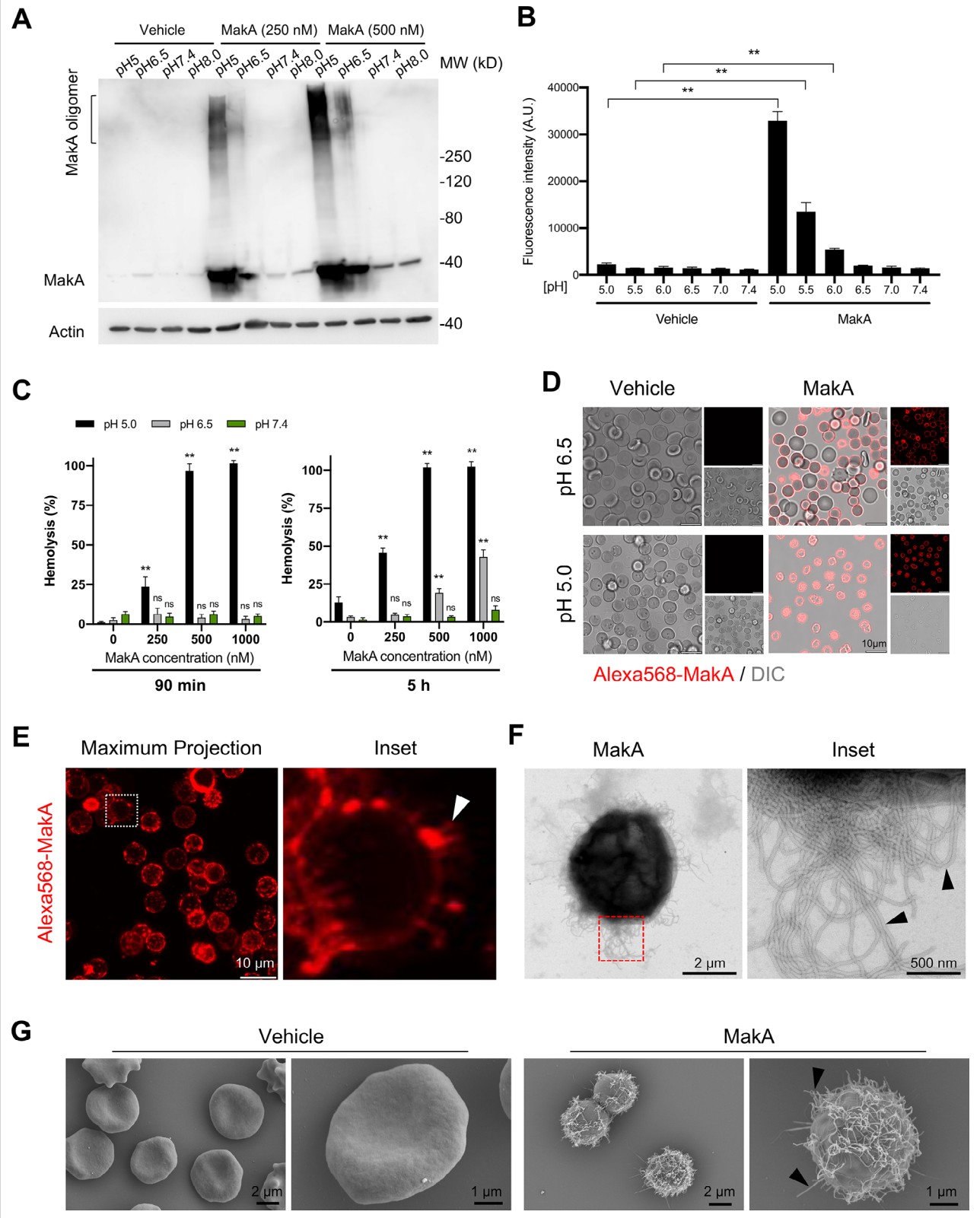

**Figure 2.** pH-dependent MakA binding to target cell membranes and induction of tubulation on erythrocytes. (**A**) Western blot analysis of HCT8 cells treated with increasing concentrations of MakA under different pH conditions for 4 hr. Data are representative of two independent experiments. Cell-bound MakA was detected with MakA-specific antibodies, and actin was used as a loading control. (**B**) HCT8 cells were treated with MakA (500 nM, 4 hr) under different pH conditions, and cell toxicity was monitored by assaying the uptake of propidium iodide. Fluorescence values for propidium

*Figure 2 continued on next page*

*Figure 2 continued*

iodide were recorded by flow cytometry. Data are representative of three independent experiments; bar graphs show the mean ± s.d. Significance was determined from biological replicates using a non-parametric t-test. **p ≤ 0.01. (**C**) Human erythrocytes suspended in a citrate buffer of different pHs were exposed to increasing concentrations of MakA for 90 min (left panel) and 5 hr (right panel). MakA-induced hemolysis of erythrocytes was normalized against erythrocytes treated with Triton X-100 (0.1%), and the data was expressed as a percentage (%). Data are representative of six readouts from two independent experiments; bar graphs show mean ± s.d. Significance was determined from biological replicates using a non-parametric t-test. **p ≤ 0.01, *p ≤ 0.05, or ns = not significant. (**D**) Human erythrocytes (0.25%) in phosphate-buffered saline (PBS) were allowed to adhere to the glass surface for 10 hr, followed by buffer exchange to citrate buffer (pH 5.0 or pH 6.5). The erythrocytes were treated with Alexa568-MakA (500 nM, 3 hr), and cell-bound MakA was detected by confocal microscopy. Scale bars, 10 μm. (**E**) The image shows a maximum z-stack projection of the human erythrocytes treated with Alexa568-MakA (pH 6.5 in citrate buffer). The white arrowhead in the right panel indicates the accumulation of Alexa568-MakA in tubular structures at the surface of erythrocytes. Scale bars, 10 μm. (**F**) Transmission electron microscopy (TEM) images of erythrocytes treated with vehicle or MakA (500 nM) for 90 min and stained with 1.5% uranyl acetate solution. Black arrowheads in the enlarged part of the image to the right indicate the presence of tubular structures on the surface of the liposome. (**G**) Scanning electron microscopy (SEM) images of erythrocytes treated with MakA (500 nM, 90 min) in citrate buffer (pH 6.5). Representative examples of imaged erythrocytes indicate that the formation of tubular structures occurred throughout the surface of MakA-treated erythrocytes. Scale bars, 2 μm.

The online version of this article includes the following source data and figure supplement(s) for figure 2:

**Source data 1.** Original Western blots for MakA and actin.

**Source data 2.** Quantification of propidium iodide-positive cells.

**Source data 3.** Quantification of hemolytic assay.

**Figure supplement 1.** pH-dependent cytotoxicity of MakA in target cells.

**Figure supplement 1—source data 1.** Quantification of trypan blue-positive cells.

**Figure supplement 1—source data 2.** Quantification of MakA-induced tubulation of erythrocytes.

## MakA induction of tubular structures on liposomes lacking other proteins

To investigate if tubulation of host membranes in response to MakA would require a specific membrane protein or receptor, we created protein- and cytosol-free liposomes using an ECLE isolated from HCT8 cells. After the addition of MakA, the liposomes were pelleted by centrifugation and the presence of MakA in either the pellet or the supernatant was detected by Western blot analysis (*Figure 3—figure supplement 1A*). The results indicated that more MakA was associated with the liposomes at pH 5.0 and 6.5 than at pH 7.4. To determine if MakA binding to ECLE under different pH conditions may lead to a confirmational change of the protein, MakA was subjected to circular dichroism (CD) spectroscopy analysis at different pH in the absence or presence of ECLE liposomes (*Figure 3A and B*). In the absence of liposomes, the CD spectra indicated a decrease in the α-helical content of the protein when in an acidic environment. This decrease in α-helical content of MakA was restored when ECLE liposomes were present, suggesting that liposomes somehow stabilized the structure of MakA (*Figure 3A and B*). TEM analysis of MakA at different pH indicated that it formed oligomeric structures in a pH-dependent manner (*Figure 3—figure supplement 1B*).

Further examination by TEM addressed whether there were morphological changes in the ECLE liposomes upon exposure to increasing concentrations of MakA at pH 6.5 (*Figure 3C* and *Figure 3—figure supplement 1C*). In a similar manner to the tubulation observed for the target cell membranes and lysosomes, MakA triggered tubulation of the ECLE liposomes in a concentration-dependent manner (*Figure 3—figure supplement 1C*). Concomitant with the assembly of tubular structures emanating from the liposomes, size of the liposomes appeared to shrink as the tubules grew up to several micrometers in length. Ultimately, the entire liposome seemed to be transformed into long tubules (*Figure 3—figure supplement 1C-E*). The tubulation of ECLE liposomes was also observed by confocal microscopy upon treatment with Alexa568-labeled MakA (*Figure 3—figure supplement 1D,E*). In the same population of small liposome particles, we also detected Alexa568-MakA-positive large lipid vesicles (5–10 μm in size, less than 1% of the entire liposome fraction). The z-stack projection suggested that the whole lipid vesicle was decorated with a bundle of fluctuating tubules (*Figure 3—figure supplement 1E*). To further assess whether or not any protein or glycolipid receptor mediated the observed membrane tubulation by MakA, liposomes were prepared from a well-defined synthetic lipid mixture (SLM); whose composition was inspired by the distribution of lipids found in the plasma membrane of HeLa cells (*Lorizate et al., 2013*). Tubulation of the SLM liposomes by MakA

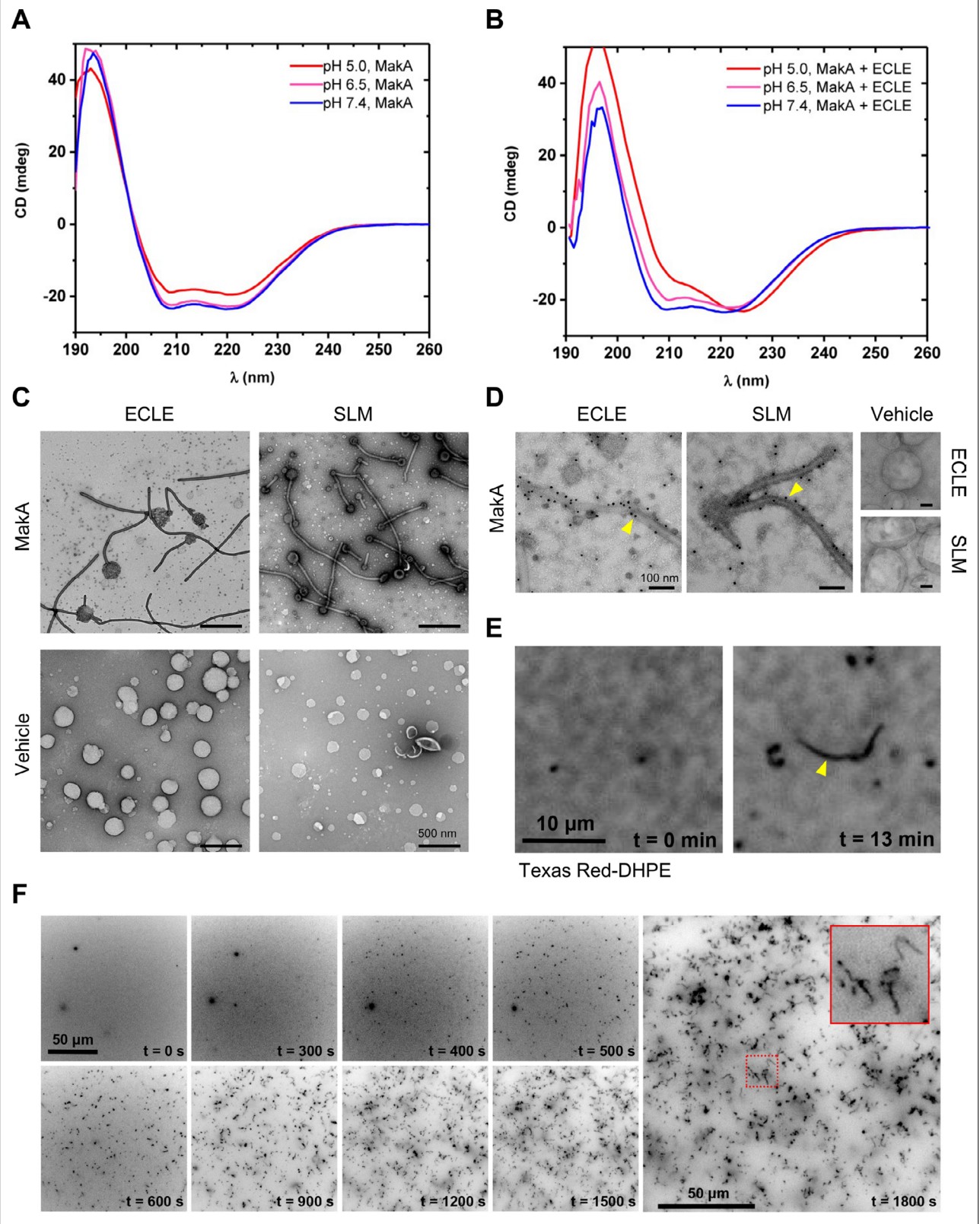

**Figure 3.** pH-dependent formation of protein-lipid tubular structures from MakA interaction with liposomes. (**A–B**) Far-UV circular dichroism (CD) spectra of native MakA of MakA bound to epithelial cell lipid extract (ECLE) liposomes under different pH conditions. CD spectra were recorded in 5 mM citrate buffer using 3 μM MakA protein. The absorption intensity measured from the control solution, containing buffer only, was subtracted to account for background absorption. (**C**) ECLE or synthetic lipid mixture (SLM) liposomes were treated at pH 6.5 with vehicle (Tris 20 mM) or MakA (3 μM)

*Figure 3 continued on next page*

*Figure 3 continued*

for 90 min and stained with a 1.5% uranyl acetate solution. Images were captured with transmission electron microscopy (TEM). White arrowheads indicate tubular structures and blue arrowheads indicate MakA oligomeric structures present in the background of liposomes. Scale bars, 200 nm. (**D**) The ECLE or SLM liposomes were treated with vehicle (Tris 20 mM) or MakA (3 μM) for 90 min and stained with a 1.5% uranyl acetate solution. MakA was detected with anti-MakA antibodies, followed by immunogold labeling and imaging by TEM. Scale bars, 200 nm. (**E**) Selected inverted grayscale images from time-lapse epifluorescence microscopy (***Video 1***) obtained after incubating supported lipid bilayers (SLBs) (prepared from SLM + TxRed liposomes) with MakA (3 μM) at pH 6.5. The fluctuating tubules (yellow arrowhead) are visible due to their TxRed-DHPE lipid content. Scale bar, 10 μm. (**F**) Selected inverted grayscale images from time-lapse epifluorescence microscopy (***Video 2***) obtained after incubating SLBs (prepared from SLM liposomes) with Alexa568-MakA (3 μM) at pH 6.5. Panels illustrate key steps during the transformation of SLBs into fluctuating tubules. The inset indicates appearance of an Alexa568-MakA-positive tubular structure. Scale bar, 50 μm.

The online version of this article includes the following source data and figure supplement(s) for figure 3:

**Source data 1.** Far-UV circular dichroism (CD) spectra values.

**Figure supplement 1.** MakA binding to epithelial cell lipid extract (ECLE) liposomes and induction of tubulation in a pH-dependent manner.

**Figure supplement 1—source data 1.** Original Western blots for MakA.

was observed by TEM (***Figure 3C***). In addition to the tubular structures, we observed a large number of well-organized, star-shaped oligomeric particles of MakA among the ECLE liposomes (***Figure 3— figure supplement 1C***). Furthermore, the presence of MakA protein in the tubular structures was evidenced by immunogold staining using MakA-specific antiserum (***Figure 3D***). By fluorescence microscopy, we were able to visualize tube growth originating from a supported lipid bilayer (SLB) prepared from SLM liposomes containing the fluorescent lipid Texas Red-DHPE, demonstrating that the tubes also contain lipids from the SLB (***Figure 3E*** and ***Video 1***).

We next investigated the kinetics of MakA protein-lipid tubulation. Using fluorescence microscopy, we found that administering Alexa568-MakA to an SLB prepared from SLM liposomes prompted a significant and highly dynamic membrane remodeling (***Figure 3F*** and ***Video 2***). Within 10 min, Alexa568-MakA binding to the SLBs resulted in formation of MakA-associated tubules of various sizes (***Figure 3F***). Based on these findings, we propose that at pH 6.5 or lower, MakA may adopt a conformation that allows the protein to insert into the lipid membrane in the form of an oligomer assembly that can lead to the formation of a tube structure. Concomitantly, the size of the vesicle appears to shrink, and the tube may grow up to several micrometers in length. Our results suggest that the

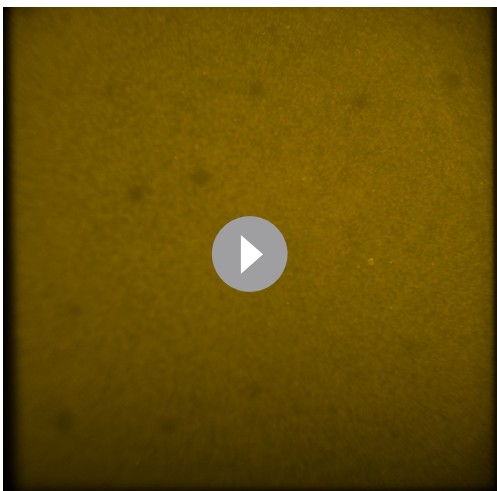

**Video 1.** Movie assembled from time-lapse fluorescence microscopy images (frame rate, 0.1 fps) obtained for TexaRed-labeled supported lipid bilayer (SLB) prepared from synthetic lipid mixture (SLM) + TxRed liposomes and treated with unlabeled MakA (3 μM) at pH 6.5. Images for generating the movie were acquired every 5 s for the duration of 20 min.

https://elifesciences.org/articles/73439/figures#video1

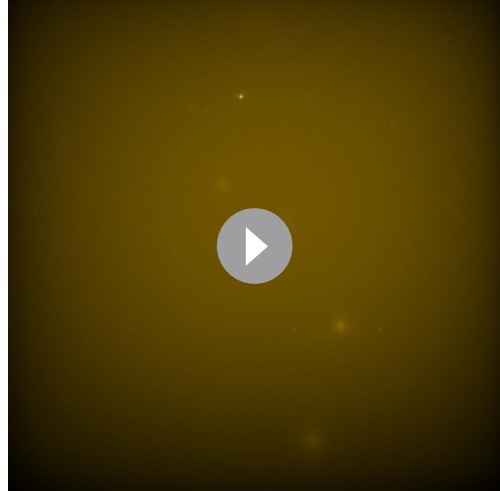

**Video 2.** Movie assembled from time-lapse fluorescence microscopy images (frame rate, 0.1 fps) obtained for supported lipid bilayer (SLB) prepared from synthetic lipid mixture (SLM) liposomes and treated with Alexa568-MakA (3 μM) at pH 6.5. Images for generating the movie were acquired every 10 s for the duration of 30 min.

https://elifesciences.org/articles/73439/figures#video2

MakA-lipid tubulation can occur without the involvement of other proteins or some specific protein receptor under the pH conditions tested.

## Structure of the MakA filament

We used helical reconstruction to solve the cryo-EM structure of a MakA filament assembled in vitro in the presence of ECLE liposomes at pH 6.5 and high protein concentrations (*Figure 4A* and *Figure 4—figure supplement 1*). An initial 2D classification allowed us to identify repetitive elements and measure a helical repeat distance of ~216 Å (*Figure 4B* and *Figure 4—figure supplement 2A*). A subsequent investigation of the layer line distances in a collapsed power spectrum of selected well-resolved class averages confirmed this distance (*Figure 4—figure supplement 2B*). Next, we performed a preliminary 3D refinement of filament segments from well-defined 2D class averages without imposing symmetry (*Figure 4—figure supplement 1*). This volume was visually inspected to deduce the helical symmetry parameters (*Figure 4—figure supplement 2C, D*). One repeating element of the right-handed spiral consists of 37 tetramers that complete five turns around the helical axis, spanning a length of 216.5 Å and a diameter of 322 Å. This arrangement results in an axial rise of 5.85 Å per subunit and a helical twist of 48.65° (*Figure 4B* and *Figure 4—figure supplement 2D*). Application of these initially calculated values with local searches in a 3D refinement further optimized symmetry parameters and resulted in a cryo-EM map at an overall resolution of 3.7 Å (*Figure 4—figure supplement 3A-D*). The obtained cryo-EM map features a well-resolved central transmembrane helix (TMH) region and a less well-resolved peripheral region (*Figure 4C*). We isolated two MakA tetramers from the segments using signal subtraction and subjected the resulting particles to 3D classification and refinement to improve the peripheral density and connectivity. The clear connectivity of the obtained density map (*Figure 4D* and *Figure 4—figure supplement 3E*) allowed for reliable secondary structure placement using the MakA soluble state crystal structure (PDB-6EZV, *Dongre et al., 2018*). High-resolution features in the helical reconstruction (*Figure 4C* and *Figure 4—figure supplement 3F*) allowed for de novo model building of structural elements in the central region. However, due to continuous rotation along the filament axis, flexibility (*Figure 4—figure supplement 1*), and the conformational difference with respect to the crystal structure, the MakA tail domain structure is less reliable and modeled with poly-alanine secondary structure elements.

MakA oligomerizes into a filamentous structure growing from or ending in membranous vesicles (*Figure 4A* and *Figure 4—figure supplement 1*). The building blocks of this filament are formed by two MakA dimers (*Figure 4B and D*) that organize into a pinecone-like architecture, spiraling around a central cavity (*Figure 4C*). From the top view along the filament axis, the helix features a propeller-like structure with a weak, annular density embedded in-between the blades formed by MakA (*Figure 4C*). This density resembles lipid tails and contains some spherical features, which could be associated with phospholipid heads, suggesting the presence of a thin phospholipid bilayer that spirals around the central cavity of the filament (*Figure 4C*). Interestingly, the annular density is located between the TMHs of MakA (*Figure 4D*), indicating that the active toxin form interacts with lipid vesicles and starts to oligomerize by internalizing membrane lipids.

## A significant conformational change is required to adopt the membrane-bound state

The basic building block of the observed protein-lipid filament is formed by four MakA subunits in a membrane-bound active conformation (*Figure 5A and B*). This conformation of MakA is significantly different from the previously reported soluble state structures resembling the inactive form (PDB-6DFP and PDB-6EZV, *Dongre et al., 2018*). In the soluble form, a C-terminal tail (res. 351–365, *Figure 5C*, purple) inactivates the predicted transmembrane domains by forming a β-tongue, consisting of three β-sheets (*Dongre et al., 2018*), that shields the hydrophobic residues from the surrounding solvent. This shielding characteristic is well described for the soluble form of the ClyA PFT family, including ClyA, Hbl-B, NheA, and AhlB (*Ganash et al., 2013*; *Kovac et al., 2016*; *Madegowda et al., 2008*; *Schubert et al., 2018*; *Wallace et al., 2000*; *Wilson et al., 2019*). MakA undergoes a structural change comparable to the opening of a Swiss army knife blade when it shifts from a soluble inactive to a membrane-bound active state, where the helix bundle of the tail region represents the handle, the transition from the tail to the neck region forms two hinges, and the β-tongue together with 4 and 5 resembles the blade that folds out (*Figure 5C*, α4 and α5; light and dark green). Additionally, the β-tongue changes its secondary structure and, together with α4 and α5/α6, forms

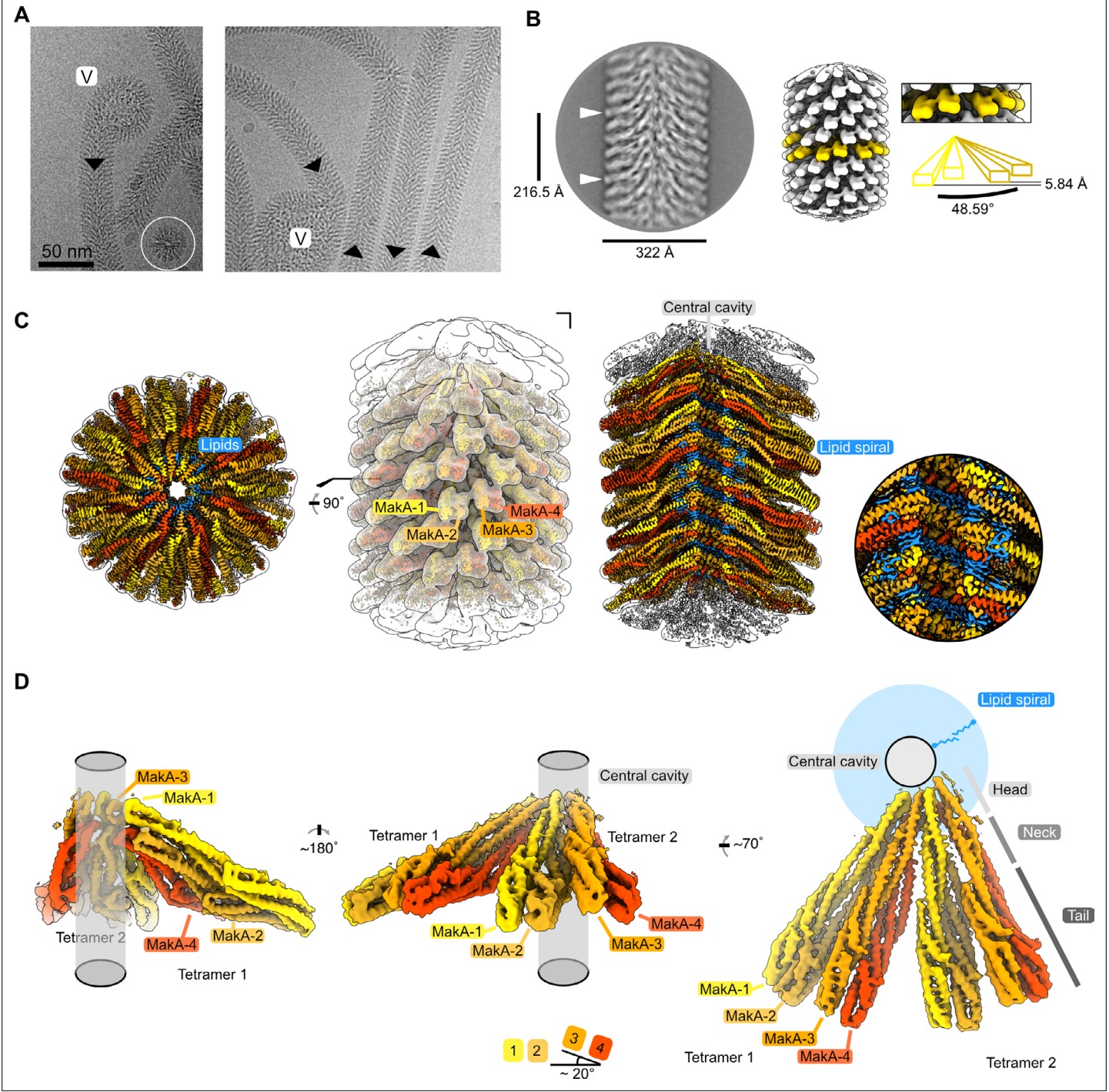

**Figure 4.** Cryo-electron microscopy (cryo-EM) structure of the membrane-bound MakA filament. (**A**) Representative cryo-EM micrograph sections showing MakA filaments emerging from or ending in a membranous vesicle (V; vesicle). The black arrows indicate the directionality of the filaments. A top view of the filamentous tube is visible in the first micrograph within the white circle. (**B**) A 2D class average with filament diameter indicated below and the repeat distance labeled on the side. An example of a visually repeating element is indicated with white arrows. The right side depicts a low-pass filtered cryo-EM volume with eight repeating subunits colored in gold next to a zoom-in of two blades. A schematic representation of the two repeating units is visualized underneath the zoom-in with a helical twist (48.59°), and a rise (5.84 Å) indicated. (**C**) Overall cryo-EM volume (EMD-13185) and slab views of the MakA filament superimposed onto a semi-transparent, white, 20 Å, low-pass filtered map. The four MakA subunits, belonging to one tetramer, are colored in shades of gold and orange-red and labeled. The different densities between the protein blades belonging to a lipid bilayer are colored in blue. (**D**) Rotationally related views of the signal of subtracted and focused-refined cryo-EM volume of two tetramers (EMD-13185-additional map 1) are shown with a schematic representation of the central cavity (transparent gray) and the lipid spiral (blue). Common structural elements of

*Figure 4 continued on next page*

*Figure 4 continued*

the alpha-cytolysin family protein-fold are indicated in gray (head, neck, and tail). The 20° rotation of the tail domain between two dimers within the asymmetric unit is shown schematically below the central panel.

The online version of this article includes the following figure supplement(s) for figure 4:

**Figure supplement 1.** Cryo-electron microscopy (cryo-EM) processing scheme.

**Figure supplement 2.** Helical symmetry determination of the MakA-filament.

**Figure supplement 3.** Global and local resolution estimation, model validation, and density fit.

**Figure supplement 4.** Comparison of the MakA and ESCRT-III subunit structure, conformation, and membrane interaction.

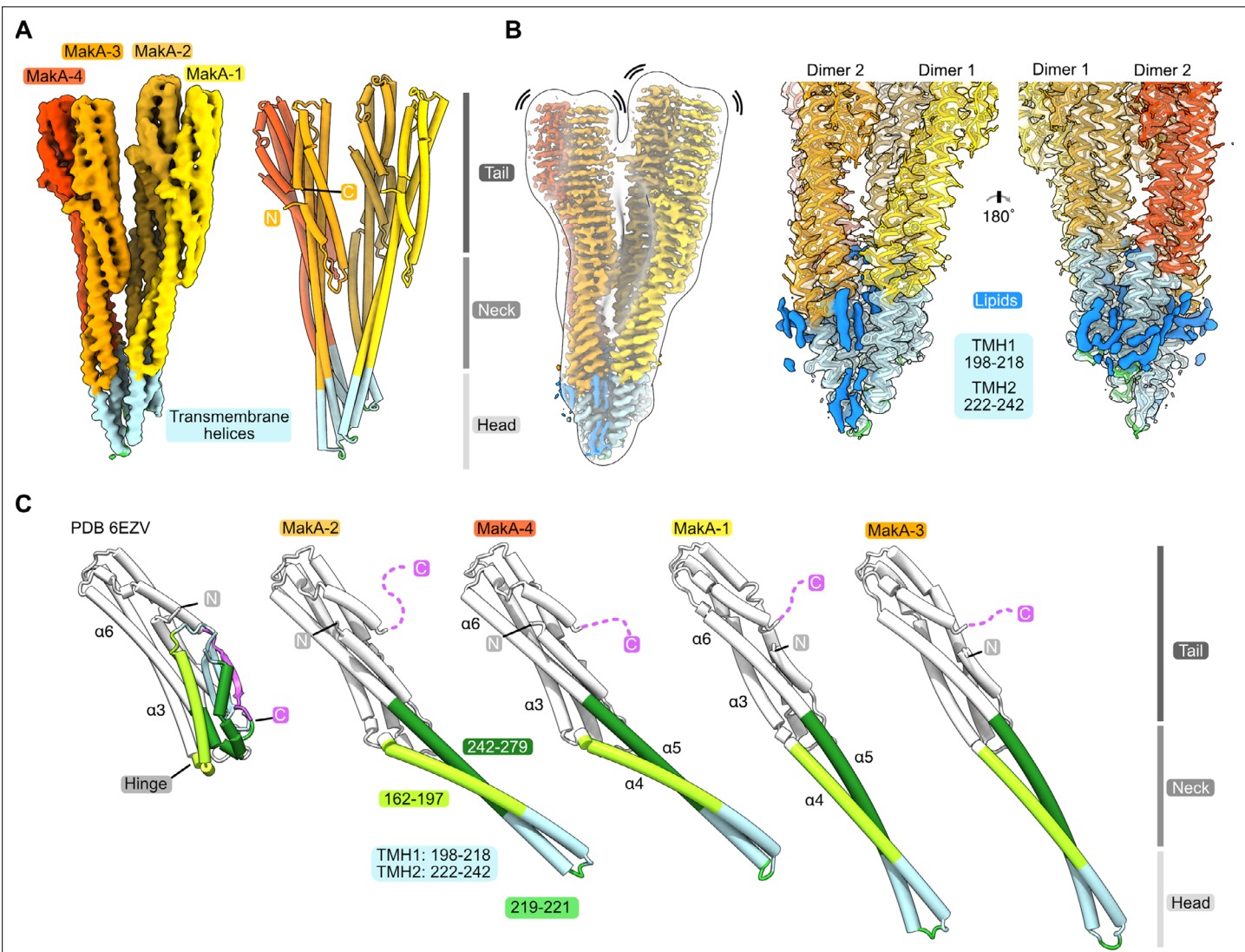

**Figure 5.** Conformational plasticity of MakA in the membrane-bound filamentous state. (**A**) The cryo-electron microscopy (cryo-EM) density of a MakA tetramer (EMD-13185-additional map 1) is shown next to a structural model. The individual protein domains (head, neck, and tail) and the visible N- and C-termini are indicated. (**B**) The cryo-EM density of a MakA tetramer, obtained by helical reconstruction (EMD-13185), is shown in isolation, colored and oriented as the structural model in (**A**). The cryo-EM volume is superimposed with a white, transparent, 20 Å, low-pass filtered volume. Additional density areas in the transmembrane helix (TMH) region, presumably belonging to lipids, are colored in blue. (**C**) The crystal structure of monomeric MakA (PDB-6EVZ, *Dongre et al., 2018*) with a retracted neck and head domain is shown next to the four individual subunits of the membrane-bound state of MakA in a cartoon representation (PDB-7P3R). All structural models were superimposed based on the tail region (in white) and displayed in the same orientation next to each other with increasing length, depicting flexing of the neck and head domain as well as the TMH.

two extended helices (*Figure 5C*). This significant conformational change leads to the formation of two TMHs and a short loop region (res. 219–221) between α4 and α5. Interestingly, despite certain similarities, all four copies of MakA adopt different conformations within the tetramer, as indicated by MakA-1 to MakA-4 (*Figures 4 and 5*). The most significant structural differences between the four MakA conformations can be observed in the neck and head domains, connected via a hinge with the tail domains (*Figure 5C*). The hinge allows for different degrees of bending (opening up of the Swiss army knife), while the neck helices α4 and α5 and the loop region (res. 219–221) display a high degree of plasticity. For the peripheral tail domain, two major conformations can be observed. In MakA-2 and MakA-4, this region superimposes well with the crystal structure, whereas helix α6 of MakA-1 and MakA-3 moves by almost 10 Å to extend the length of α5. While the stretched MakA states have shorter N-terminal helices than the kinked MakA forms, the C-terminal β-strand (res. 351–365), which covers the TMH in the soluble state, is disordered in all four subunits.

## Discussion

Recent studies have revealed how different bacterial species, notably *B. cereus*, *A. hydrophila*, *S. marcescens*, and *V. cholerae*, have the capability to produce structurally similar tripartite protein complexes that assemble on host cell membranes as pore structures that are cytolytic (*Churchill-Angus et al., 2021*; *Nadeem et al., 2021c*; *Sastalla et al., 2013*; *Wilson et al., 2019*). In the case of *V. cholerae,* it has become evident that the MakA component of the tripartite complex, when presented alone to mammalian cells, is effectively internalized and accumulates on endolysosomal membranes, leading to induction of autophagy and apoptotic cell death (*Corkery et al., 2021*; *Nadeem et al., 2021a*; *Nadeem et al., 2021b*). Moreover, MakA can modulate host cell autophagy in a pH-dependent manner (*Corkery et al., 2021*). Herein, we show that MakA is able to produce a novel protein-lipid polymeric superstructure at a low pH (6.5 or below) that perturbs host cell membranes.

Several prokaryotic proteins are known to polymerize in the presence of a matrix such as a lipid membrane or DNA. This type of assembly is referred to as a collaborative filament (*Ghosal and Löwe, 2015*). In the presence of lipid membranes, collaborative filaments are assembled by a bridging protein and lipid on the membrane in a sequence-specific manner and by sensing membrane curvature (*Ghosal and Löwe, 2015*). It was demonstrated that membrane-mediated clustering of Shiga toxin molecules and the formation of tubular membrane invaginations are essential steps in the clathrin-independent Shiga toxin uptake process (*Pezeshkian et al., 2016*). In earlier studies, pH-dependent membrane insertion and increased cytotoxic activity were demonstrated for different bacterial toxins, including anthrax toxin from *Bacillus anthracis* (*Koehler and Collier, 1991*), diphtheria toxin from *Corynebacterium diphtheriae* (*Rodnin et al., 2016*), VacA from *Helicobacter pylori* (*Pagliaccia et al., 2000*), perfringolysin from *Clostridium perfringens* (*Nelson et al., 2008*), and listeriolysin O from *Listeria monocytogenes* (*Schuerch et al., 2005*). Recent research on the pre- and pore forms of a mammalian PFT, the protein perforin 2 (mPFN2), provides interesting insights into the pore generation process. It was demonstrated that pre-pore-to-pore transformation occurs at an acidic pH, which is accomplished by a 180° rotation of the membrane-attacking domain and β-hairpin P2 domain with respect to one another, allowing membrane insertion to take place (*Ni et al., 2020*). Similar to these other toxins, a decrease in pH facilitated the MakA structural change comparable to the opening of a Swiss army knife blade when shifting from a soluble to a membrane-bound active state that interacts with the lipid membrane. The pH-induced structural change potentiated the MakA-induced cytotoxic effect on the target cell under the conditions tested. Consistent with these results, we recently found that MakA co-localizes with β-catenin, actin, and the phosphatidic acid biosensor, PASS, in filopodia-rich structures (*Nadeem et al., 2021a*; *Nadeem et al., 2021b*).

Our in vitro analysis of the low-pH-induced interaction between MakA and host membranes included purified lysosomes, cultured epithelial cells, red blood cells, and liposome models. In all model systems, we observed that low pH enhanced the activity of MakA, characterized by a striking tubulation of both purified lysosomes and liposomes prepared from ECLE. Furthermore, it was found that MakA formed tubular structures on liposomes independent of any specific protein receptor or energy-generating molecules, that is, ATP or GTP, suggesting that at low pH, the structural change and insertion of MakA itself was sufficient to trigger tubulation at the membrane surface. In contrast to some other PFTs that need cholesterol for oligomerization (*Gellings and McGee, 2018*; *Rosado et al., 2008*; *Sathyanarayana et al., 2018*), MakA could induce tubulation on liposomes obtained from lipid sources essentially lacking cholesterol, that is, *E. coli* and *C. elegans*, showing that the pH-dependent tubulation of MakA can occur in the absence of cholesterol in the membrane. Bacterial

cell membranes are typically devoid of cholesterol (*Fiertel and Klein, 1959*), while the membranes of *C. elegans* are mostly composed of glycerophospholipids and sphingolipids with trace quantities of cholesterol (*Watts and Ristow, 2017*). In addition, we observed that the presence of cholesterol was not required for the oligomerization of MakA in solution (*Figure 3—figure supplement 1B*).

Importantly, our in vitro studies do not indicate that a particular membrane topology per se is required for the MakA-induced tube formation. The data indicate that MakA can interact with membrane lipids representing either side or topology of a membrane bilayer and thereby induce tubulation when the pH is acidic. This membrane interaction by MakA thereby appears different from, for example, Shiga toxin and CT, which insert into the membrane and cause inward-directed tubulations of artificial lipid membranes since toxin binding induces negative curvature of the plasma membrane (*Chinnapen et al., 2012*; *Römer et al., 2007*; *Safouane et al., 2010*). The CD spectrometry and TEM-negative staining data indicate that under acidic conditions MakA undergoes a conformational change, presumably adopting a structure that is promoting membrane interaction and formation of oligomeric assemblies (*Figure 2A* and *Figure 1—figure supplement 1B*).

MakA evidently drives the rapid growth of tubules toward the extracellular space, as shown with red blood cells (*Figure 2E-G*). We propose that the appearance of tubular structures in response to MakA was a direct consequence of MakA insertion into the membrane, which appeared to create the conditions for generating a positive curvature, as described previously for protein-lipid complexes and multi-anchoring polymers (*Campelo, 2007*; *Nadeem et al., 2015*).

MakA and ESCRT-III proteins seem to have some similarities in terms of the major refolding of the subunit with ESCRT-III proteins. Both proteins tubulate membranes ultrastructurally and, as observed for MakA, human ESCRT-III protein CHMP1B/IST1 generates a positive membrane curvature (*Nguyen et al., 2020*). However, there are major differences between the proteins in terms of membrane interaction and oligomeric assembly: Through positively charged phospholipid-interacting regions in helix 1 (*Figure 4—figure supplement 4*, shown in red), CHMP1B proteins associate with and coat a membrane externally, resulting in a stable, continuous, and intact membrane. MakA, on the other hand, has transmembrane domains that integrate into and span the lipid bilayer. When the membrane is disassembled and linearized, a lipid spiral forms around the filament axis, together with the MakA subunits. This implies membrane instability, which is consistent with the proposed cytotoxicity of MakA as opposed to ESCRT-III-regulated membrane remodeling and fission. Moreover, reduced ionic strength activates ESCRT-III (*McCullough et al., 2015*), whereas the trigger for MakA appears to be pH dependent. CHMP1B in its open, active conformation needs its binding partner, IST1, to co-polymerize and promote membrane tubulation (*McCullough et al., 2015*), while MakA oligomerizes into a helical filament on its own in vitro (by forming pairs of dimers). It should be noted that when MakA is interacting in vitro with its other binding partners from *V. cholerae*, the structurally related MakB and MakE proteins, it acts as a component in a tripartite complex, forming oligomeric pore structures in membranes at neutral pH (*Nadeem et al., 2021c*). The observed structural heterogeneity within the dimer pair of MakA shown at acidic pH in the present study presumably reflects its ability to engage in heteromeric interactions with the other two Mak proteins. Furthermore, oligomerized CHMP1B and MakA have dramatically different protein-protein interactions: MakA creates pairs of dimers that are arrayed like a single propellor blade, while CHMP1B binds to four successive CHMP1B monomers that are almost parallel packed next to each other, resulting in a low-fence-like structure (*McCullough et al., 2015*). Our findings allowed us to propose a model for MakA oligomer assembly and its implications for pore formation. Within the oligomerized MakA filament, the TMHs line the inner cavity, interacting with lipids that are intercalated both within the dimer interface and between the dimers in the helical spiral (*Figures 4C and 5B*). In the context of PFT systems, the initial insertion of one toxin component was described for XaxAB (*Schubert et al., 2018*), YaxAB (*Brauning et al., 2018*), and suggested for AhlC (*Wilson et al., 2019*). Considering that the TMHs of the MakA-tetramers harbor a lipid bilayer, it could be assumed that MakA initiates membrane insertion in a manner similar to its role in pore formation under neutral pH conditions when part of the tripartite MakA/B/E complex (*Nadeem et al., 2021c*). We hypothesize that under low-pH conditions, MakA transitions from inactive to the active stretched conformation and penetrates the membrane as a monomer. In the membrane-bound state, the interaction of the head and tail domains of two monomers might lead to MakA dimer formation and subsequent oligomerization that ultimately lead to membrane tubulation.

The pronounced structural difference between two subunits forming a dimer in the absence of the tripartite complex interaction partner, that is, MakB or MakE, would reflect how MakA's plasticity nevertheless can structurally mimic the expected structure when involved in the tripartite cytolysin. First, one completely stretched monomer dimerizes with a monomer with a pronounced elbow-like kink from α3 to α4 in the transition from the tail to the neck region (*Figure 5C*). Subsequently, the two dimers form a tetramer with the stretched and kinked MakA states associating with each other, respectively. The subsequent tetramerization followed by oligomerization does not result in pore formation, but in helical growth and lipids being sheared off from vesicles and cell membranes at low pH. This process, which does not occur in the tripartite cytotoxin scenario under neutral pH conditions, depletes membranous structures of lipids and potentially causes cell lysis in a manner quite different from that of a bona fide α-PFT cytolysin complex. Our findings with the *V. cholerae* MakA protein reveal an unexpected capability and remarkable mode of action of an individual α-PFT toxin subunit.

The actual physiological role(s) in terms of function for the bacteria, and the pH-dependent MakA activity, remain(s) to be elucidated. A feasible hypothesis emerges when we consider the fact that the intestinal lumen of *C. elegans* worms is comparable to the endolysosomal compartment of human cells in terms of the low pH levels. The pH of the intestinal lumen of a living *C. elegans* varies from 5.96 in the pharynx to 3.59 in the lower intestine (*Chauhan et al., 2013*). In using *C. elegans* as a predatory model organism feeding on MakA-producing bacteria it will be of interest to determine if the pH-dependent activity of MakA actually plays a direct role in the toxic effect on the worms. Investigation of MakA's pathophysiological effects and molecular interactions with intestinal tissue and cell membranes in the worms should reveal if analogous tube-like structures are formed.

From the perspective of protein engineering and nanotube formation, we suggest that the present findings on formation of MakA lipid/protein spiral structures have potential to be useful in fields of protein/membrane engineering and synthetic biology.

## Materials and methods

**Key resources table**

| Reagent type (species) or resource | Designation | Source or reference | Identifiers | Additional information |
|---|---|---|---|---|
| Cell line (*Homo sapiens*) | Large intestine; Colon | ATCC | HCT 8 (RRID:CVCL_2478) | Cell line maintained in RPMI-1640 |
| Cell line (*Homo sapiens*) | Large intestine; Colon | ATCC | Caco-2 (RRID:CVCL_0025) | Cell line maintained in RPMI-1640 |
| Cell line (*Homo sapiens*) | Large intestine; Colon | ATCC | HCT 116 (RRID:CVCL_0291) | Cell line maintained in RPMI-1640 |
| Antibody | Anti-MakA (Rabbit polyclonal) | GeneCust | PO# AB190007 | WB (1:20,000) |
| Antibody | Anti-LAMP1 (Rabbit polyclonal) | Cell Signaling | Cat# 9091S RRID:AB_2687579 | WB (1:1000) |
| Antibody | Anti-β-actin (Mouse monoclonal) | Sigma-Aldrich | Cat# A2228 RRID:AB_476697 | WB (1:5000) |
| Antibody | Goat anti-Rabbit IgG HRP conjugated (polyclonal) | Agrisera | Cat# AS09602 RRID:AB_1966902 | WB (1:5000) |
| Antibody | Rabbit anti-Mouse IgG HRP conjugated (polyclonal) | Dako | Cat# P0260 RRID:AB_2636929 | WB (1:5000) |
| Commercial assay or kit | Clarity Western ECL reagent | Bio-Rad | Cat# 170–5061 | Western blot reagent |
| Commercial assay or kit | Isolation of lysosomes from HCT8 or Caco-2 cells | Abcam | Cat# ab234047 | |
| Commercial assay or kit | SuperSignal West femto maximum sensitivity substrate | Thermo Scientific | Cat# 34096 | Western blot reagent |
| Transfected construct (human) | CellLight Lysosomes-GFP | Invitrogen | Cat# C10507 | GFP-LAMP1 transfection reagent for lysosome labeling |

*Continued on next page*

*Continued*

| Reagent type (species) or resource | Designation | Source or reference | Identifiers | Additional information |
| --- | --- | --- | --- | --- |
| Software, algorithm | GraphPad Prism | GraphPad Prism | RRID:SCR_002798 | |
| Software, algorithm | Fiji | Fiji | RRID:SCR_002285 | |
| Other | Alexa Fluor 568 | Thermo Fisher Scientific | Cat# A10238 | Alexa568-MakA labeling |
| Other | Propidium Iodide | BD Pharmingen | Cat# 66211E | For flow cytometry (0.5 µg/mL) |
| Other | Hoechst 33342 | Thermo Scientific | Cat# 62249 | (1–2 µM) |

## Chemicals and lipids

Chloroform, formaldehyde, methanol, sodium citrate, Tween 20, Triton X-100, and Fluoromount were from Sigma (St Louis, MO). Hoechst 33342 and Lysotracker were from Thermo Fisher Scientific (Waltham, MA). Propidium iodide was from BD Biosciences (San Jose, CA). Protease inhibitor and phosphatase inhibitor, phosSTOP were from Roche (Roche AB, Solna, Sweden). All lipids were purchased from Avanti Polar Lipids (Alabaster, AL). Lipids: 1-palmitoyl-2-oleoyl-glycero-3-phosphocholine (POPS), 1,2-dioleoyl-*sn*-glycero-3-phosphoethanolamine (DOPE), 1-palmitoyl-2-oleoyl-*sn*-glycero-3-phospho-L-serine (POPS), Sphingomyelin from Porcine Brain (SM), Cholesterol from Ovine (Chol), L-α-phosphatidylinositol-4,5-bisphosphate from Porcine Brain (PIP2), and *N*-palmitoyl-sphingosine-1-{succinyl[methoxy(polyethylene glycol)5,000]} (PEG5Kce). Lyophilized PIP2 lipids were dissolved in a mixture of chloroform:methanol (2:1) to a concentration of 1 mg/mL. Next, they were protonated by addition of 0.5 µL of 1 M HCl to 100 µg of PIP2, kept at room temperature (RT) for 15 min and dried with nitrogen gas. The dried lipid was redissolved in chloroform: methanol (3:1) mixture to 1 mg/mL followed by drying again. Finally, the 100 µg of PIP2 was redissolved in 100% chloroform to 1 mg/mL and stored at –20°C until used for liposome production.

## Mammalian cell culture

Caco-2 (ATCC), HCT8 (ATCC), and HCT116 (ATCC) cells were cultured in RPMI-1640 medium (Sigma-Aldrich) supplemented with 10% fetal bovine serum (FBS), 1% penicillin/streptomycin, and non-essential amino acids. Cells were cultured at 37°C, 5% $CO_2$, and 90% humidity in 96-well plates overnight (for cell viability assays), coverslip bottom eight-well chamber slides (for confocal and spinning disc confocal microscopy), 75 $cm^2$ flasks (for lysosome isolation), 24-well plates (for flow cytometry), and six-well plates (for Western blot analysis).

## Antibodies

Anti-MakA antiserum produced by GeneCust (1:20,000 dilution), anti-LAMP1 antibody (#9091) purchased from Cell Signaling (1:1000 dilution), and anti-beta-actin antibody (A2228) purchased from Sigma-Aldrich (1:5000 dilution) were used in this study.

## Cloning and purification of MakA

Cloning, overexpression, and purification of MakA have been previously reported (*Dongre et al., 2018*). Alexa Fluor568 labeling of MakA was performed using an Alexa Fluor568 protein labeling kit (Thermo Fisher) according to the manufacturer's instructions.

## Isolation and treatment of intact epithelial cell lysosomes

HCT8 cells were grown overnight in RPMI-1640 complete media (~pH 7.2). The following day, cells were treated with MakA (250 nM, 18 hr). At the end of treatment, lysosomes were purified from vehicle- or MakA-treated HCT8 cells using the Lysosome Isolation Kit (ab234047, Abcam), according to the manufacturer's instructions.

For the lysosome pull-down assay, intact lysosomes freshly isolated from HCT8 cells were diluted in three times their volume of freshly prepared binding buffer (120 mM sodium citrate, pH 5.0, pH 6.5, or pH 7.0), followed by incubation with MakA (20 µg/mL) at 37°C (60 min). These MakA-lysosome complexes were centrifuged at 21,000× *g* (30 min, RT). Pellets were washed in the respective binding

buffer and resuspended in 2× Laemmli buffer. Samples were run on an SDS-PAGE, and after electrophoresis, the samples were transferred to a nitrocellulose membrane. A Western blot analysis was performed using anti-MakA antiserum (1:20,000 dilution, overnight at 4°C) that was detected with HRP-conjugated goat anti-rabbit secondary antibodies. Detection of LAMP1, using anti-LAMP1 antibodies, was used as an internal loading control for lysosome pull-down experiments. The membranes were developed with a chemiluminescence reagent (Bio-Rad). Images were acquired using an ImageQuant LAS 4000 instrument and processed using ImageJ-FIJI distribution (*Schindelin et al., 2012*).

For confocal microscopy, intact lysosomes diluted in three times their volume of freshly prepared binding buffer (120 mM sodium citrate, pH 5.0, pH 6.5, or pH 7.0) were exposed to Alexa568-MakA (1 µM). To facilitate the binding, Alexa568-MakA and lysosomes were incubated in a 37°C incubator for 60 min. At the end of treatment, samples were visualized by a Leica SP8 inverted confocal system (Leica Microsystems) equipped with an HC PL APO 63×/1.40 oil immersion lens. Images were captured using LasX software (Leica Microsystems) and processed using ImageJ-FIJI distribution (*Schindelin et al., 2012*).

## Live-cell spinning disk microscopy

Live-cell experiments were conducted in phenol-red-free IMDM media adjusted to pH 5.0 supplemented with 10% FBS and 1 mM sodium pyruvate (Thermo Fisher Scientific) at 37°C in 5% $CO_2$. Alexa568-MakA (500 nM) was added to HCT8 cells, and images were recorded every 1 min during a period of 120 min using a 63× lens and Zeiss Spinning Disk Confocal controlled by the ZEN interface (RRID:SCR_013672) with an Axio Observer Z1 inverted microscope, equipped with a CSU-X1A 5000 Spinning Disk Unit and an EMCCD camera iXon Ultra from ANDOR. Images were processed with Zeiss ZEN Lite and ImageJ-FIJI distribution (*Schindelin et al., 2012*).

## Immunofluorescence

Lysosomal tubulation was investigated by treating Caco-2 cells with Alexa568-MakA (250 nM, 18 hr) in IMDM complete media (pH 7.2). After treatment, cells were subsequently counterstained for lysotracker (200 nM, 30 min) and Hoechst 33342 (2 µM, 30 min).

For confocal microscopy, HCT8 cells were loaded with the nuclear staining marker Hoechst 33342 (2 µM, 30 min) and exposed to Alexa568-MakA (500 nM) in different pH-adjusted IMDM complete media for 4 hr at 37°C in a 5% $CO_2$ incubator. Cells were visualized live using a Leica SP8 inverted confocal system (Leica Microsystems) equipped with an HC PL APO 63×/1.40 oil immersion lens. Images were captured using the LasX (Leica Microsystems) and processed using ImageJ-FIJI distribution (*Schindelin et al., 2012*).

For the propidium iodide uptake experiment, HCT8 cells were treated with MakA (500 nM, 4 hr) in IMDM complete media (pH 5.0 or pH 7.4), followed by adding propidium iodide (0.5 µg/mL, 30 min). Fluorescence and bright-field images were captured with a fluorescence microscope (Nikon, Eclipse Ti). Images were processed using the NIS-Elements (Nikon) and ImageJ-FIJI distribution (*Schindelin et al., 2012*).

For Alexa568-MakA binding to erythrocytes, freshly prepared human erythrocytes (0.25% in PBS) were loaded into an eight-well chamber slide (µ-Slide, ibidi). Cells were allowed to adhere to the glass surface for 10 hr, followed by buffer exchange to citrate buffer (pH 5.0, pH 6.5, or pH 7.4). The erythrocytes were exposed to Alexa568-MakA (500 nM) for 3 hr at 37°C in a 5% $CO_2$ incubator. Cells were visualized using a Leica SP8 inverted confocal system (Leica Microsystems) equipped with an HC PL APO 63×/1.40 oil immersion lens. The maximum z-stack projection of the human erythrocytes treated with Alexa568-MakA (pH 6.5 in citrate buffer) was constructed using Leica LasX Software.

## Cell toxicity assay

HCT8, Caco-2, and HCT116 cells were treated with increasing concentrations of MakA at a given pH at the indicated time point (4 or 24 hr). Cells were detached with trypsin at the end of treatment, and viability was determined by counting trypan blue-positive cells. The percentage of trypan blue-positive cells indicate cell death.

For flow cytometry experiments and confocal microscopy, HCT8 cells were grown on a 24-well plate (8 × 10⁴/well, Tecan Group Ltd) overnight in IMDM complete media (pH 7.2). The following day, cells were treated with vehicle (Tris 20 mM) or MakA (500 nM, 4 hr) in media adjusted to a given pH.

At the end of treatment, cells were incubated with propidium iodide (0.5 µg/mL) at 37°C for 30 min. Cellular uptake of propidium iodide in vehicle- or MakA-treated cells was investigated by flow cytometry or confocal microscopy. For flow cytometry experiments, cellular uptake of propidium iodide was quantified and presented as mean fluorescence intensity for the gated cells.

### Human erythrocyte hemolysis assay

Freshly prepared human erythrocytes (0.25%) in citrate buffer (120 mM sodium citrate, pH adjusted to 5.0, 6.5, or 7.4) were added to a 96-well plate. The erythrocytes were treated with increasing concentrations of MakA at two different time points (90 min and 5 hr) at 37°C in a 5% $CO_2$ incubator. After centrifugation (500× $g$), the supernatants were monitored spectrophotometrically for released hemoglobin by measuring absorbance at 545 nm to indicate red blood cell lysis. MakA-induced hemolysis of erythrocytes was normalized against erythrocytes treated with Triton X-100 (0.1%). Data were expressed as a percentage.

### Scanning electron microscopy

Freshly prepared human erythrocytes (0.25%) were treated with MakA (500 nM, 90 min) in citrate buffer (120 mM, pH 6.5). Samples were fixed with a fixative (1% glutaraldehyde +0.1 M CaCo buffer +3 mM $MgCl_2$) in the microwave and washed twice with buffer (0.1 M CaCo buffer +2.5% sucrose +3 mM $MgCl_2$). They were sedimented onto poly-L-lysine-coated coverslips for 1 hr and subsequently dehydrated in a series of graded ethanol solutions in the microwave. The samples were then dried to a critical point (Leica EM300). Subsequently, samples were coated with 2 nm of platinum (Quorum Q150T ES). Samples were imaged with field-emission scanning electron microscopy (Carl Zeiss Merlin) using an in-chamber (ETD) secondary electron detector at an accelerating voltage of 5 kV and a probe current of 150 pA.

### Extraction of epithelial cell lipids for liposome binding assays

Lipids were extracted by the Folch method (*Folch et al., 1957*) from 10 × 150 cm² confluent flasks of HCT8 cells. Briefly, the HCT8 cell lipid extracts, dissolved in chloroform, were dried to a thin film under a nitrogen stream. The dried lipid yield was 12 mg. The lipid film (5 or 10 mg/mL) was hydrated in HEPES buffer (10 mM HEPES, 150 mM NaCl, pH 7.4), citrate buffer (120 mM citrate buffer, pH 6.5), or citrate acid buffer (20 mM citric acid, 50 mM KCl, 0.1 mM EDTA, pH 4.5). The lipid suspension was extruded through polycarbonate membranes (0.1 µm) using the Avanti Mini-Extruder (Avanti Polar Lipids, Alabaster, AL).

The liposome pull-down assay was performed as previously described (*Julkowska et al., 2013*; *Nadeem et al., 2021a*). Briefly, the liposome suspension was diluted in five times its volume of freshly prepared binding buffer (120 mM sodium citrate, pH 5.0, pH 6.5, or pH 7.4), followed by centrifugation at 21,000× $g$ for 30 min at RT. The liposome pellet was resuspended in binding buffer followed by incubation with MakA (20 µg/mL). The liposome-protein mixtures were incubated at 37°C (60 min), followed by centrifugation at 21,000× $g$ at RT (30 min). To reduce the background, pellets were washed in the respective binding buffer two to three times. The resulting sample was loaded onto the SDS-PAGE, transferred to a nitrocellulose membrane, and subjected to Western blot analysis using anti-MakA antiserum (1:10,000 dilution, overnight at 4°C). The MakA antibodies were detected with HRP-conjugated goat anti-rabbit secondary antibodies. The membranes were developed with a chemiluminescence reagent (Bio-Rad). Images were acquired using ImageQuant LAS 4000 instrument and processed using ImageJ-FIJI distribution (*Schindelin et al., 2012*).

### CD spectroscopy

Far-UV CD analysis of MakA protein or MakA and ECLE liposome complexes was performed using Jasco J-720 Spectropolarimeter (Japan) at 25°C. Briefly, MakA (3 µM) alone or MakA (3 µM) and ECLE (5 mg/mL) were incubated overnight at 25°C in citrate buffer (5 mM) with varying pH. The spectra were recorded between 195 and 260 nm using a 2 s response time, a 1 mm cuvette path length, and a 2 nm bandwidth. Data of an average of five repeated scans were used for graphical presentation and analyses.

### Liposome preparation

Liposomes containing 0.5 mol% PEG5Kce, 5 mol% PIP2, 10 mol% SM, 10 mol% POPS, 15 mol% DOPE, 20 mol% Chol, and 39.5 mol% POPC (referred to herein as SLM liposomes) were prepared

using the lipid film hydration and extrusion method. SLM+ TxRed liposomes were created using the same protocol as above except for the addition of 1 mol% TxRed-DHPE and the corresponding reduction of POPC content to 38.5 mol%. The individual lipids dissolved in chloroform were mixed together, dried under nitrogen, and then stored in a vacuum for a minimum of 1 hr. The dried lipid film was then rehydrated using a pre-heated citrate-potassium buffer at pH 4.5 (20 mM citric acid, 50 mM KCl, pH 4.5, 40°C) to a lipid concentration of 1 mg/mL. The solution was then extruded at ~40°C 11 times through a polycarbonate membrane with a 100 nm pore size using an Avanti mini extruder. The liposomes were stored at 4°C until used.

## Fluorescence microscopy of SLBs

SLBs were formed on glass coverslips. Coverslips were cleaned by boiling in 7× detergent (MP Biochemicals) for 2 hr followed by extensive rinsing in 18 MΩ water and blow drying with nitrogen. Clean coverslips were fitted with poly(dimethylsiloxane) sheets containing 10 μm holes to create glass-bottomed wells. SLBs were made by adding 10 μL of 0.1 mg/mL of either SLM or SLM + TxRed liposomes to each well and incubating the wells at 37°C for 30 min before rinsing the wells with citrate-potassium buffer to remove excess liposomes. Wells were then extensively rinsed with citrate buffer at pH 6.5 (120 mM sodium citrate) prior to protein addition. Either Alexa568-MakA or unlabeled MakA diluted in citrate buffer at pH 6.5 was then added to a well to reach a final concentration of 3 μM. The SLB surface was monitored using a Nikon Eclipse Ti2-E inverted epifluorescence microscope equipped with a 60× objective multi-band pass filter cube 86,012 v2 DAPI/FITC/TxRed/Cy5 (Nikon Corp.), Prime 95B sCMOS camera (Teledyne Photometrics), and Spectra III light source (Lumencor).

## Western blot analysis

For Western blot analysis, HCT8 cells were grown on a six-well slide ($3 \times 10^5$/well, Thermo Scientific), overnight. The pH of IMDM cell culture media supplemented with 10% FBS and 1% penicillin/streptomycin was adjusted to either 5.0, 6.5, 7.4, or 8.0, followed by treatment with an increasing concentration of MakA for 4 hr. Cells were rinsed with ice-cold PBS to remove unbound MakA and lysed in ice-cold NP-40 cell lysis buffer (20 mM Tris-HCl pH 8, 0.25% Nonidet P-40, 10% glycerol, 0.5 mM EDTA, 300 mM KCl, 0.5 mM EGTA, 1× phosSTOP, and protease inhibitor cocktail from Roche). For Western blot analysis of purified lysosomes, HCT8 or Caco-2 cells were grown in a 75 cm² flask ($5 \times 10^6$ cells/flask), overnight. Cells were treated with vehicle or MakA (250 nM, 18 hr) and lysosomes were purified using the Lysosome Isolation Kit (ab234047, Abcam), according to the manufacturer's instructions. Purified lysosomes from HCT8 and Caco-2 cells were crosslinked with glutaraldehyde (0.05%) for 10 min at 37°C followed by the addition of stop solution Tris-HCl (200 mM, pH 6.8). Subsequently, sample buffer was added to the vehicle- or MakA-treated lysosomes with or without crosslinking. After mixing with sample buffer, cell lysates or lysosomes were boiled for 5 min and separated by SDS-PAGE. The proteins were transferred to a nitrocellulose membrane and blocked with 5% skim milk in 0.1% PBST (RT, 1 hr). The membranes were incubated with the respective primary antibodies in 5% skim milk (4°C, overnight). After washing with PBST (0.1%), membranes were incubated with HRP-conjugated secondary antibodies in 5% skim milk (RT, 1 hr). The membranes were developed with chemiluminescence reagent, Clarity Western ECL substrate (Bio-Rad) or Super-Signal West femto maximum sensitivity substrate (Thermo Scientific). Images were acquired using ImageQuant LAS 4000 instrument and processed using ImageJ-FIJI distribution (*Schindelin et al., 2012*).

## Transmission electron microscopy

Negative staining for lysosomes or liposomes was performed on glow discharged copper grids (300 mesh) coated with a thin carbon film (Ted Pella, Redding, CA). After adding 3 μL sample to the grids, they were washed twice with MQ water and stained with 1.5% uranyl acetate solution (EMS, Hatfield, PA), followed by MQ water washing. Grids were examined with Talos L120C, operating at 120 kV. Transmission electron micrographs (TEM) were acquired with a Ceta 16M CCD camera using TEM Image & Analysis software ver. 4.17 (FEI, Eindhoven, The Netherlands).

## Cryo-EM sample preparation and data collection

The ECLE liposomes (10 mg/mL) were incubated with MakA (30 µM) in binding buffer (120 mM sodium citrate, pH 6.5) for 60 min at 37°C. Quantifoil 2/1–200 grids were glow discharged before the addition of 3 µL protein-liposome mixture. Grids were then flash-frozen in liquid ethane using an FEI Vitrobot (Thermo Fisher Scientific). Data collection was performed at the Umeå University Core Facility for Electron Microscopy (UCEM) on a Titan Krios (Thermo Fisher Scientific), operating at 300 kV and equipped with a Gatan K2 BioQuantum direct electron detector (Gatan, Inc). Images were acquired using EPU (Thermo Fisher Scientific). A total of 2476 movies, each with 40 frames over a total dose of 43 e-/$Å^2$, and a 0.75–2.5 µm defocus range at a 1.042 Å pixel size were collected.

## Cryo-EM data processing and helical reconstruction

The MotionCorr implementation of RELION-3.1 was used for drift correction and dose weighting of the micrographs (*Zivanov et al., 2018*). The contrast transfer function (CTF) was determined using CTFFIND-4.1.14 (*Rohou and Grigorieff, 2015*), and empty or micrographs with poor CTF fits or low ice quality were removed after manual inspection, which reduced the total to 1351 micrographs (*Figure 4—figure supplement 1*). Helical reconstruction tools in RELION-3.1 (*He and Scheres, 2017*) were used for subsequent image processing. Filaments were selected manually using the helix picker in RELION-3.1 with an outcome of 13,784 picked start and endpoints. First, segments were extracted between the picked start and endpoints as 2× binned data using a box size of 437 Å (210 pixels, 2.084 Å/px) with an inter-box distance (IBD) of 23 Å, resulting in 195'809 segments, which were subjected to 2D classification with 80 classes and a spherical mask of 360 Å. Classes displaying a straight filament with high-resolution features (152'961 segments) were refined without symmetry using a featureless cylinder (diameter 320 Å) generated with the helix toolbox in RELION-3.1 (*Figure 4—figure supplement 1*). In parallel, a 2D classification was performed by extracting 65'241 segments from 2× binned data, applying a box size of 646 Å (310 px, 2.084 Å/px), an IBD of 62 Å, and a spherical mask of 580 Å.

The volume, refined without symmetry, was used to determine the helical symmetry parameters. First, diameter and repeat distance were visually analyzed in this volume (*Figure 4B* and *Figure 4—figure supplement 4A*). Further, the repeat distance was confirmed from a corresponding summed power spectrum and by measuring the layer-line distance-1 in collapsed power spectra obtained using SPRING-0.86 (*Desfosses et al., 2014*) or BSOFT (*Heymann and Belnap, 2007*) and averaged using RELION-3.1 (*Figure 4—figure supplement 2B*). Next, to determine the helical twist and rise, the number of turns and subunits per repeat were counted from the initial reconstructed model (*Figure 4—figure supplement 2C,D*), and the handedness of the reconstruction was, after the subsequent high-resolution refinement described below, confirmed using the MakA crystal structure (*Dongre et al., 2018*).

For the final reconstruction, 95'603 segments were extracted unbinned with 460-pixel boxes (479 Å) and an IBD of 46.56 Å. Subsequent 2D classification with a 432 Å spherical mask resulted in 65'485 segments that were 3D refined using a featureless cylinder as a template and a spherical mask of 360 Å. Local symmetry searches were performed to narrow down the helical symmetry by refining helical twist (48–50°) and rise (5.4–6 Å), yielding a map with an overall 4.1 Å resolution. Subsequent Bayesian polishing improved the resolution to 3.8 Å, and estimation of anisotropic magnification and CTF refinement resulted in a final map with an overall resolution of 3.7 Å (*Figure 4—figure supplement 3*). As the peripheral region was less well resolved, a subsection of the structure was isolated via signal subtraction, centered in a 260-pixel box and subjected to 3D classification without symmetry and local searches with an increasing sampling rate from 3.7°, 1.8°, and 0.9° angles. From the resulting three classes, further refinement of class 2 (56.8%) yielded a map of the two tetramers in isolation at an overall resolution of 4.1 Å with improved peripheral density (*Figure 4—figure supplement 4A,B*, blue branch, and *Figure 4—figure supplement 3D,E*). As the resulting three classes showed different conformations of the MakA head region, we examined whether these conformations exist across the volume. Two subsequent 3D classifications into five classes, without image alignment but with local symmetry searches, first with all classes, then with the top class from the first run, showed a normal distribution of angles, ranging from 48.48° to 48.68° suggesting continuous motion/rotation along the filament axis, which is most pronounced in the tail region (*Figure 4—figure supplement 1*, blue branch, lower right).

## Model building, refinement, and validation

To obtain an initial model of the tail domain, the MakA crystal structure (PDB-6EZV; *Dongre et al., 2018*) was rigid-body docked into the density map using Chimera (*Pettersen et al., 2004*) and Coot (*Emsley and Cowtan, 2004*). Regions where the density/model fit was poor (no density, difference in conformation) were trimmed. This included the C-terminal tail (res. 351–365) and the central region (res. ~ 160–260). Elements in the tail domain with poor density fit were rigid-body docked. The central region of the protein, which includes the neck and head, was built de novo in Coot (*Emsley and Cowtan, 2004*). The model was first refined against the asymmetric map of the two tetramers in

**Table 1.** Cryo-electron microscopy (cryo-EM) data collection, refinement, and model statistics.

| | MakA helical reconstruction | MakA, tetramers in isolation |
|---|---|---|
| | **EMD-13185; PDB-7P3R** | **EMD-13185- additional map 1** |
| **Data collection and processing** | | |
| Voltage (kV) | 300 | 300 |
| Pixel size (Å) | 1.042 | 1.042 |
| Electron exposure (e-/Å[Beecher and Wong, 1997]) | 43 | 43 |
| Defocus range (μm) | 0.7–2.5 | 0.7–2.5 |
| Frames | 40 | 40 |
| Symmetry imposed | | C1 |
| Helical twist (°) | 48.59 | – |
| Helical rise (Å) | 5.84 | – |
| Initial particle images | 95'603 | 95'603 |
| Final particle images | 65'485 | 37'876 |
| Resolution (Å) | 3.7 | 4.1 |
| FSC threshold | 0.143 | 0.143 |
| Map sharpening B-factor (Å[Beecher and Wong, 1997]) | –99.9 | –117 |
| **Refinement** | | |
| Initial model used | 6EZV | 6EZV |
| Model composition | | |
| Non-hydrogen atoms | 7'738 | |
| Protein residues | 1'338 | |
| R.m.s deviations | | |
| Bond length (Å) | 0.0069 | |
| Angles (°) | 1.17 | |
| Validation | | |
| MolProbity score | 1.08 | |
| Clashscore | 1.88 | |
| Poor rotamers (%) | 0.28 | |
| Ramachandran | | |
| Favored (%) | 97.29 | |
| Allowed (%) | 2.71 | |
| Outliers (%) | 0.00 | |

isolation (4.1 Å) using phenix.real_space_refine (version 1.14–3260) (*Adams et al., 2010*). Next, this model was rigid-body docked into the 3.7 Å helical map, two neighboring placeholder molecules were symmetry expanded to provide interaction interfaces, and refined with secondary structure restraints. The final model contains four MakA subunits with trimmed sidechains in the tail domain (N-terminus to 159, 281 to C-terminus, *Figure 4—figure supplement 3E,F*). To validate the final model, all atomic coordinates were displaced randomly by 0.5 Å, refined against half map 1, followed by calculating the Fourier shell correlation coefficient of the resulting refined model and half map 1 or half map 2 (*Brown et al., 2015*). Model statistics are presented in *Figure 4* and *Table 1*.

## Map and model visualization

Structure analyses and preparation of the figures were performed using PyMOL (Schrödinger) or UCSF ChimeraX (*Goddard et al., 2018*).

## Statistical analysis

The result from replicates is presented as mean ± s.e.m. or mean ± s.d. The statistical significance of different groups was determined by Student's t-tests (two-tailed, unpaired) or one-way ANOVA using Microsoft Excel or GraphPad Prism. *$p \leq 0.05$, **$p \leq 0.01$, ns = not significant.

## Acknowledgements

We thank Dr Irina Gutsche for valuable comments and advice on the cryo-EM analysis. This work was supported by grants from the Swedish Research Council (No. 2018–02914 to SNW; No. 2016–05009 to KP; No. 2019–01720 to BEU; No. 2016–06963 to GG), The Swedish Cancer Society (No. 2017–419 and No. 2020–711 to SNW), The Kempe Foundations (No. JCK-1728 to SNW; No. SMK-1756.2 and No. SMK-1553 to KP; No. JCK-1724 and No. SMK-1961 to BEU), and the Faculty of Medicine, Umeå University (Strategic Research Grant 2019–2021 to SNW). MB was supported by the Knut and Alice Wallenberg Foundation. JB acknowledges funding from the Swedish Research Council (2019–02011), the European Research Council (ERC Starting Grant PolTube 948655), the SciLifeLab National Fellows program, and MIMS. We acknowledge the Protein Expression and Purification facility (PEP) at Umeå University for construct design and cloning. We acknowledge the facilities and technical assistance of the Umeå Core Facility Electron Microscopy (UCEM) and the Biochemical Imaging Center (BICU), Umeå University, a part of the National Microscopy Infrastructure NMI (VR-RFI 201600968 and VR-RFI 2019–00217). The cryo-EM data were collected by UCEM, which is a node of the Swedish National Cryo-EM Facility, funded by the Knut and Alice Wallenberg Foundation, Erling-Persson Family Foundation, The Kempe Foundations, SciLifeLab, Stockholm University, and Umeå University.

## Additional information

### Competing interests

Aftab Nadeem, Karina Persson, Bernt Eric Uhlin, Sun Nyunt Wai: S.N.W., B.E.U., A.N., and K.P. wish to make the disclosure that we are named inventors in a PCT application (Vibrio cholerae protein for use against cancer) published under No. WO 2021/071419. This does not alter our adherence to eLife policies on sharing data and materials. The other authors declare that no competing interests exist.

### Funding

| Funder | Grant reference number | Author |
|---|---|---|
| Vetenskapsrådet | No. 2018-02914 | Sun Nyunt Wai |
| Vetenskapsrådet | No. 2016-05009 | Karina Persson |
| Vetenskapsrådet | No. 2019-01720 | Bernt Eric Uhlin |
| Vetenskapsrådet | No. 2016-06963 | Gerhard Gröbner |
| Swedish Cancer Foundation | No. 2017-419 | Sun Nyunt Wai |

| Funder | Grant reference number | Author |
|---|---|---|
| Swedish Cancer Foundation | No. 2020-711 | Sun Nyunt Wai |
| Kempestiftelserna | No. JCK-1728 | Sun Nyunt Wai |
| Kempestiftelserna | No. SMK-1756.2 and No. SMK-1553 | Karina Persson |
| Kempestiftelserna | No. JCK-1724 and No. SMK-1961 | Bernt Eric Uhlin |
| Umeå Universitet | Strategic Research Grant 2019-2021 | Sun Nyunt Wai |
| Vetenskapsrådet | No. 2019-02011 | Jonas Barandun |
| European Research Council | ERC Starting Grant PolTube 948655 | Jonas Barandun |

The funders had no role in study design, data collection and interpretation, or the decision to submit the work for publication.

## Author contributions

Aftab Nadeem, Conceptualization, Data curation, Formal analysis, Investigation, Methodology, Software, Validation, Visualization, Writing – original draft, Writing – review and editing; Alexandra Berg, Hudson Pace, Athar Alam, Karina Persson, Marta Bally, Jonas Barandun, Data curation, Formal analysis, Investigation, Methodology, Software, Validation, Visualization, Writing – original draft, Writing – review and editing; Eric Toh, Formal analysis, Investigation, Methodology, Validation, Visualization, Writing – review and editing; Jörgen Ådén, Data curation, Formal analysis, Investigation, Methodology, Software, Validation, Visualization, Writing – review and editing; Nikola Zlatkov, Gerhard Gröbner, Data curation, Formal analysis, Investigation, Methodology, Validation, Visualization, Writing – review and editing; Si Lhyam Myint, Data curation, Formal analysis, Investigation, Methodology, Validation, Visualization; Anders Sjöstedt, Formal analysis, Resources, Visualization, Writing – review and editing; Bernt Eric Uhlin, Sun Nyunt Wai, Conceptualization, Data curation, Formal analysis, Funding acquisition, Investigation, Methodology, Project administration, Resources, Software, Supervision, Validation, Visualization, Writing – original draft, Writing – review and editing

## Author ORCIDs

Aftab Nadeem http://orcid.org/0000-0002-1439-6216
Alexandra Berg http://orcid.org/0000-0003-3609-2878
Hudson Pace http://orcid.org/0000-0001-5116-2577
Athar Alam http://orcid.org/0000-0001-8773-7598
Eric Toh http://orcid.org/0000-0002-0103-0696
Jörgen Ådén http://orcid.org/0000-0002-4480-1219
Nikola Zlatkov http://orcid.org/0000-0003-3318-9084
Si Lhyam Myint http://orcid.org/0000-0001-5384-3691
Karina Persson http://orcid.org/0000-0003-0807-0348
Gerhard Gröbner http://orcid.org/0000-0001-7380-8797
Anders Sjöstedt http://orcid.org/0000-0002-0768-8405
Marta Bally http://orcid.org/0000-0002-5865-8302
Jonas Barandun http://orcid.org/0000-0003-2971-8190
Bernt Eric Uhlin http://orcid.org/0000-0002-2991-8072
Sun Nyunt Wai http://orcid.org/0000-0003-4793-4671

## Decision letter and Author response

Decision letter https://doi.org/10.7554/eLife.73439.sa1
Author response https://doi.org/10.7554/eLife.73439.sa2

# Additional files

## Supplementary files
• Transparent reporting form

## Data availability

The cryo-EM density maps have been deposited in the EM Data Bank with accession code EMD-13185 (MakA helical reconstruction) and EMD-13185-additional map 1 (two tetramers refined in isolation). Coordinates have been deposited in the Protein Data Bank under accession code PDB-7P3R.

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
