## [Editor Report]

This paper provides some remarkable insights about the pore-forming toxin MakA, from *V. cholerae*, showing how it alone can bind to membranes and form regular tubes. Given the general interest in pore-forming toxins in terms of both understanding bacterial pathogenesis and designing new therapeutics, the paper should be of broad interest.

---

## [Decision Letter]

**Decision letter after peer review:**

Thank you for submitting your article "Protein-lipid interaction at low pH induces oligomerisation of the MakA cytotoxin from *Vibrio cholerae*" for consideration by *eLife*. Your article has been reviewed by 3 peer reviewers, including Edward H Egelman as Reviewing Editor and Reviewer #1, and the evaluation has been overseen by Richard Aldrich as the Senior Editor. The following individuals involved in review of your submission have agreed to reveal their identity: Adam Frost (Reviewer #2); Ana J Garcia-Saez (Reviewer #3).

Essential revisions:

1) The topology of the structure raises a concern as it appears to be the opposite of what would occur in a cellular context. This needs to be addressed.

2) The authors should clarify whether these experimental conditions for the addition of MakA to the extracellular medium at acidic pH would have any physiological relevance in the context of the actual host/pathogen interaction

3) The map:model FSC cannot be believed, or at least it cannot be understood why it is so much better than the map:map FSC.

4) There appears to be some similarity to the major refolding of ESCRT-III proteins. This should be either discussed or dismissed.

*Reviewer #1 (Recommendations for the authors):*

There are some general problems with the approach used for helical symmetry determination. While it certainly appears that the correct symmetry has been found for the inner core, it is entirely possible that the outer region may have a reduced symmetry (such as by dimerization) that leads to a lower resolution when the core symmetry is imposed. A simple way to check this would be by generating an averaged power spectrum from the segments. Does this power spectrum only show the helical symmetry of the core, or are their additional layer lines that cannot be explained by the core? This is quite different from what has been done, which relies on power spectra from 2D class averages. As has been shown, the averaged power spectrum is not the same as the power spectrum of an averaged image. While what I suggest may sound far-fetched, it has actually been shown for bacterial flagellar filaments that the core can have a higher symmetry than the outer domains, which can form dimers in some species.

There is something very strange about the map:model FSC, since the apparent resolution is ~ 3.0 Å. How is this possible?

There appear to be some similarities in terms of the major refolding of the subunit with ESCRT-III proteins. This should be discussed. Is there potential homology?

*Reviewer #2 (Recommendations for the authors):*

1) Can the authors distinguish possible mechanisms for how acidic pH triggers the MakA change, the "opening of a Swiss army knife blade"? Which residues near the C-terminal tail, hinges, or β-tongue may be pH sensitive?

2) Figure 3A. These binding isoforms have unusual response profiles that make them challenging to interpret. This protein's binding and unbinding from membranes is not a simple first-order process: MakA polymerizes upon the membrane. Do the authors think the upward slope reflects ongoing assembly? How are the liposomes anchored to the L1 chip? How does the reported binding constant account for polymerization (I don't think it does, so this is a Kapparent of 49nm?). On balance, these data add little to the story.

3) The high-resolution features recovered in the helical reconstruction indicate that the symmetry imposed is "correct" -- at least to the FSC-estimated resolution of the map. However, the procedure described for determining the symmetry from a 2D class average is not accurate and should not be perpetuated. Specifically, the authors wrote "Diameter and repeat distance were visually analyzed and measured in a representative 2D class average in RELION-3.1 (Figure 4b and Supplementary Figure 5a). Additionally, the repeat distance was calculated from the corresponding collapsed power spectrum (layer-line distance-1) in SPRING-0.68 (ref. 51 715 ) (Supplementary Figure 5b)." The power spectrum of a class average is NOT the same as an averaged power spectrum and can lead to error. Please visit this discussion and this paper (https://discuss.cryosparc.com/t/averaged-power-spectrum/7017 and https://elifesciences.org/articles/04969).

4) Supp Figure 6b seems to indicate that the map-to-model correspondence at FSC=0.5 is ~3A? I suspect this is wrong because the halfmap FSC correspondence does not exceed 3.7A in the best case.

*Reviewer #3 (Recommendations for the authors):*

– The lysosome tubulation induced by MakA in cells requires the opposite membrane topology to the conditions used by the authors in this study. The authors should reconcile these contradictory observations.

– Only when pH was acidic in the cell culture medium did the authors detect plasma membrane tubulation in cells, similar to the one they find in model membranes in terms of protein/membrane topology. The authors should clarify whether these experimental conditions for the addition of MakA to the extracellular medium at acidic pH would have any physiological relevance in the context of the actual host/pathogen interaction. The authors should also address whether membrane tubulation of the plasma membrane induced by MakA induce cell death of the HCT8 cells.

– The authors should explore in more detail the lipid dependency of the membrane alterations induced by MakA, not just using pure PC as a control. They could incorporate for example negatively charged lipids, PIPs, cholesterol and sphingomyelin using the Biacore assay.

– The authors should clarify if lipids form a continuous bilayer along the tube or are they laterally separated by protein transmembrane domains. If the membrane is not continuous, the authors should address whether the permeability is affected, which would be important in the context of the tubulation observed in cells.

---

## [Author Response]

Essential revisions:1) The topology of the structure raises a concern as it appears to be the opposite of what would occur in a cellular context. This needs to be addressed.

This concern has been addressed as described in the point-by-point response below to Reviewer #3 as follows:

The results from our in vitro studies do not indicate that a particular membrane topology *per se* is required for the MakA induced tube formation.

On the contrary, we interpret our data to indicate that MakA can interact with membrane lipids representing either side or topology of a membrane bilayer and thereby induce tubulation when the pH is acidic.

The CD spectrometry results, and TEM negative staining images indicate that under acidic conditions, MakA undergoes conformational change, presumably adopting a structure that promotes membrane interaction and the formation of oligomeric assemblies in the presence of membrane lipids. (Figure 2A and Figure 2—figure supplement 1B).

Similarly, we infer that a pH-induced conformational change and oligomerization of MakA in lysosomes may lead to lysosomal tubulation (Figure 1A). Western blot analysis of lysosomes isolated from MakA treated HCT8 and Caco-2 cells detected formation of dimeric, tetrameric and oligomeric MakA complexes in addition to the monomer (Figure 1A and Figure 1—figure supplement 1B), which is consistent with the notion that MakA oligomerization inside lysosomes may be responsible for lysosomal tubulation.

2) The authors should clarify whether these experimental conditions for the addition of MakA to the extracellular medium at acidic pH would have any physiological relevance in the context of the actual host/pathogen interaction

As the initial point of interest, we recently discovered that MakA accumulates in the acidic endolysosomal compartment and causes lysosomal dysfunction (PMID: 33720402). Based on these observations, we hypothesized that pH may regulate the activity of MakA. In additional experiments we exposed the target cells to MakA under different pH conditions. Consistent with our hypothesis, we observed dramatic increase in the activity of MakA when the pH of the extracellular media was acidic (Figure 1 and Figure 2).

In the Discussion section of our amended manuscript, we have pointed out that the actual physiological role(s) in terms of function for the bacteria of the pH-dependent MakA activity remains to be elucidated.

Nevertheless, we also point out a feasible case where the acidic pH effect on MakA might have physiological relevance in the context of a host/bacterium interaction. With the use of *C. elegans*, which served as a predatory model organism, we earlier proved that MakA is a fitness factor for *Vibrio cholerae* and is essential for the killing effect on worms (PMID: 30271941). We now consider the fact that the pH of the intestinal lumen of a live *C. elegans* ranges from 5.96 in the pharynx to 3.59 in the lower intestine (PMID: 23668893). In light of our present findings, we suggest that it will be of interest to investigate the pathophysiological effects of MakA and its molecular interactions in the intestine of *C. elegans*.

3) The map:model FSC cannot be believed, or at least it cannot be understood why it is so much better than the map:map FSC.

This issue has been addressed, and updated model-validation FSC curves have been included in the revised manuscript (as also described in the point-by-point answers to Reviewer #1 and Reviewer #2) as follows:

To exclude tightly bound lipid densities and close adjacent proteins, we employed a mask that was, in retrospect, too tight.

For model validation, we repeated the FSC calculation using a more generous mask based on a 20 Å lowpass filtered tetramer at a threshold that covers the entire volume, and extended the original mask by 4 pixels. Figure 4—figure supplement 3 has been replaced by this model validation curve, which includes updated FSC curves with labeled resolution for FSC cut-off values of 0.5 (4.1 Å) or 0.143 (3.6 Å).

4) There appears to be some similarity to the major refolding of ESCRT-III proteins. This should be either discussed or dismissed.

We find that a structural resemblance to the ESCRT-III proteins exists only at the ultrastructural level, and not at the molecular level. There is no obvious sequence homology or structural similarity between the MakA and ESCORT-III proteins. Furthermore, the modes of membrane interaction appears different between the two. These aspects are now highlighted in the Discussion section by referring to the recent studies by Nguyen et al., (2020). To illustrate the clarified points, we included an additional supplementary figure that displays the apparent differences between the pore-forming toxin MakA's unique oligomeric form and the ESCRT-III proteins, (Figure 4—figure supplement 4).

Reviewer #1 (Recommendations for the authors):There are some general problems with the approach used for helical symmetry determination. While it certainly appears that the correct symmetry has been found for the inner core, it is entirely possible that the outer region may have a reduced symmetry (such as by dimerization) that leads to a lower resolution when the core symmetry is imposed. A simple way to check this would be by generating an averaged power spectrum from the segments. Does this power spectrum only show the helical symmetry of the core, or are their additional layer lines that cannot be explained by the core? This is quite different from what has been done, which relies on power spectra from 2D class averages. As has been shown, the averaged power spectrum is not the same as the power spectrum of an averaged image. While what I suggest may sound far-fetched, it has actually been shown for bacterial flagellar filaments that the core can have a higher symmetry than the outer domains, which can form dimers in some species.

We thank Reviewer #1 for the thorough reading of our manuscript and for providing very constructive input. We are addressing each point raised in the comments below and believe that the manuscript has improved substantially.

Reviewer #1 raised a valid point regarding the possible differing symmetry of the inner and outer filament cores. To investigate this possibility, we calculated the summed power-spectra (PS) from 1331 selected segments as well as from 271 projections of the final volume and compared them. All power spectra agree well with each other, including the PS of the 2D class average, and no additional layer lines are visible (Figure 4—figure supplement 2B). Furthermore, we have confirmed the symmetry parameters by performing broad global searches of the helical twist and rise in CryoSPARC (Figure 4—figure supplement 2F).

Based on our analysis (Figure 4—figure supplement 1 and 2), we assume that continuous motion along the filament axis, together with the dynamic nature of the solvent-exposed dimer, contributes to the reduced resolution in the peripheral region rather than imposing a symmetry that only describes the core. Our assumption is based on a number of factors:

– Movement / flexing along the filament axis: A 3D classification without image alignment but with local symmetry searches (Figure 4—figure supplement 1, blue branch, lower right) shows a normal distribution of angles with a major particle population at a twist of 48.57 degree. Reclassification of this population again results in normal distributed particles. We expect such behaviour for continuous states along the filament axis.

– Local conformational differences and flexibility of the solvent exposed tail: The isolation of two MakA tetramers from the segments using signal subtraction followed by 3D classification and refinement (without imposing symmetry) allowed us to improve the peripheral density and connectivity. At the same time, this analysis visualized significant local conformational differences and structural heterogeneity in the tail region rather than specific states.

– Filament and interdimer wobbling due to membrane fluidity: The filament is built up like a spring with lipids in between the turns. The thin lipid bilayer might act like a lubricant for the protein spiral. The non-uniform nature of the lipid mixture introduces additional heterogeneity in thickness and membrane-behavior. We expect the MakA proteins embedded in a thin lipid bilayer with no protein/protein interactions between turns and only minimal interactions between asymmetric subunits to be dynamic.

– Suboptimal interactions within the dimer interface: Recent in vitro studies show that MakA is part of a tripartite pore-forming toxin complex consisting of three structurally homologous proteins (MakA, MakB, and MakE). The homodimer could represent a suboptimal structural state, with reduced stability or steric problems, and therefore potentially does not form a perfect and stable dimer as it might form with MakB, MakE, or in a trimer with both. However, such heterodimers remain to be characterized at the ultrastructural level.

Taken together, these points suggest that the less well resolved tail region of MakA is a result of motion. To clarify the procedure used to determine the helical symmetry parameters, we have adjusted the text in the method section and the legend of Figure 4—figure supplement 2**.** Furthermore, for full transparency, we have uploaded the raw dataset to the public repository EMPIAR under accession code EMPIAR-10869

There is something very strange about the map:model FSC, since the apparent resolution is ~ 3.0 Å. How is this possible?

To exclude tightly bound lipid densities and close adjacent proteins, we employed a mask that was, in retrospect, too tight.

For model validation, we repeated the FSC calculation using a more generous mask based on a 20 Å lowpass filtered tetramer at a threshold that covers the entire volume, and extended the original mask by 4 pixels. Figure 4—figure supplement 2B has been replaced by this model validation curve, which includes updated FSC curves with labeled resolution for FSC cut-off values of 0.5 (4.1 Å) or 0.143 (3.6 Å).

There appear to be some similarities in terms of the major refolding of the subunit with ESCRT-III proteins. This should be discussed. Is there potential homology?

Both proteins promote tubulation of membranes ultrastructurally, and, as observed for MakA, the human ESCORT-III protein CHMP1B/IST1 induces a positive membrane curvature. However, the proteins exhibit considerable differences in terms of membrane interaction and oligomeric assembly: Through positively charged phospholipid-interacting regions in helix 1 (Figure 4—figure supplement 4**,** indicated in red), CHMP1B proteins associate with, and coat, a membrane externally, resulting in a stable continuous and intact membrane. MakA in contrast, possesses transmembrane domains that integrate into and span the lipid bilayer. When the membrane is disassembled and linearized, a lipid spiral around the filament axis forms, together with the MakA subunits. This indicates destabilization of the membrane, which is consistent with the proposed cytotoxicity of MakA, as opposed to the membrane remodeling and fission regulated by ESCRT-III.

MakA and CHMP1B have no obvious structural or sequence homology at the molecular level. However, certain similarities can be observed regarding active and inactive monomer conformations. Both proteins exist in a closed, soluble, inactive state in which the C-terminus shields the membrane-interacting region from the surrounding solvent (McCullough et al., Science, 2015; Shim et al., Traffic, 2007; Tang et. al, *eLife*, 2016). The inactive monomers transition into an open, active, insoluble conformation, in which the proteins’ secondary structure changes from loop (CHMP1B) or a β sheet (MakA) to a membrane-associated helix in CHMP1B (McCullough et al., Science, 2015) and to the MakA transmembrane helices which are exposed to the environment. Activation of ESCRT-III is achieved by reduction of ionic strength (McCullough et al., Science, 2015), whereas the trigger for MakA seems to be pH dependent.

The open, active conformation of CHMP1B, requires its binding partner, IST1, to co-polymerize and promote membrane tubulation (McCullough et al., Science, 2015), whereas MakA oligomerizes on its own into a helical filament in vitro by forming pairs of dimers. However, as stated in the Discussion section, it should be noted that when MakA is interacting in vitro with its other binding partners from *V. cholerae*, the structurally related MakB and MakE proteins, it acts as a component in a tripartite complex forming oligomeric pore structures in membranes at neutral pH (Nadeem et al., 2021). The observed structural heterogeneity within the dimer pair of MakA shown in the present study at acidic pH is presumably reflecting its ability to engage in heteromeric interactions with the other two Mak proteins.

Moreover, the protein-protein interactions of oligomerized CHMP1B and MakA are very different: While one CHMP1B molecule binds to four subsequent CHMP1B monomers that are almost parallel packed next to each other resulting in a low-fence-like structure (McCullough et al., Science, 2015), MakA forms pairs of dimers, which are organized in the shape of a single propellor blade.

To clarify the similarities and differences between MakA and ESCRT-III, we have included an additional figure (Figure 4—figure supplement 3) for comparative purposes and added a paragraph for clarification to the Discussion.

Reviewer #2 (Recommendations for the authors):1) Can the authors distinguish possible mechanisms for how acidic pH triggers the MakA change, the "opening of a Swiss army knife blade"? Which residues near the C-terminal tail, hinges, or β-tongue may be pH sensitive?

The question raised by Reviewer #2 regarding possible mechanisms for how acidic pH triggers the MakA change cannot really be answered at this stage. We initiated different approaches, including attempts by site-directed mutagenesis, to probe which residues may be pH sensitive. However, as of yet it is premature for us to make any conclusion or suggestion regarding possible mechanisms. We hope to be able to characterize the actual mechanism(s) in some future contribution.

2) Figure 3A. These binding isoforms have unusual response profiles that make them challenging to interpret. This protein's binding and unbinding from membranes is not a simple first-order process: MakA polymerizes upon the membrane. Do the authors think the upward slope reflects ongoing assembly? How are the liposomes anchored to the L1 chip? How does the reported binding constant account for polymerization (I don't think it does, so this is a Kapparent of 49nm?). On balance, these data add little to the story.

We agree with Reviewer #2 that the results reported in Figure 3A are challenging to interpret for the very reasons given. As it is questionable if these Surface Plasmon Resonance data can add anything to the story, we decided to omit the experiment, and we have removed this figure.

3) The high-resolution features recovered in the helical reconstruction indicate that the symmetry imposed is "correct" -- at least to the FSC-estimated resolution of the map. However, the procedure described for determining the symmetry from a 2D class average is not accurate and should not be perpetuated. Specifically, the authors wrote "Diameter and repeat distance were visually analyzed and measured in a representative 2D class average in RELION-3.1 (Figure 4b and Supplementary Figure 5a). Additionally, the repeat distance was calculated from the corresponding collapsed power spectrum (layer-line distance-1) in SPRING-0.68 (ref. 51 715 ) (Supplementary Figure 5b)." The power spectrum of a class average is NOT the same as an averaged power spectrum and can lead to error. Please visit this discussion and this paper (https://discuss.cryosparc.com/t/averaged-power-spectrum/7017 and https://elifesciences.org/articles/04969).

The helical symmetry was not derived from a 2D class average or its power spectrum. As described in the methods section, particles were refined without imposing symmetry against a cylindrical volume as a reference. We then analyzed the resulting volume to obtain the symmetry parameters. The analyses of the 2D class average and its PS were used to verify obtained parameters. Furthermore, we did not utilize the PS for indexing. We have added a summed power spectrum and a PS of the final model to Figure 4—figure supplement 2. Additionally, as described in our response to Reviewer #1, we clarified the method section and caption of Figure 4—figure supplement 1.

4) Supp Figure 6b seems to indicate that the map-to-model correspondence at FSC=0.5 is ~3A? I suspect this is wrong because the halfmap FSC correspondence does not exceed 3.7A in the best case.

Reviewer #1 raised a similar question, and we repeat our response here:

To exclude tightly bound lipid densities and close adjacent proteins, we employed a mask that was, in retrospect, too tight.

For model validation, we repeated the FSC calculation using a more generous mask based on a 20 Å lowpass filtered tetramer at a threshold that covers the entire volume, and extended the original mask by 4 pixels. Figure 4—figure supplement 2B has been replaced by this model validation curve, which includes updated FSC curves with labeled resolution for FSC cut-off values of 0.5 (4.1 Å) or 0.143 (3.6 Å).

Reviewer #3 (Recommendations for the authors):– The lysosome tubulation induced by MakA in cells requires the opposite membrane topology to the conditions used by the authors in this study. The authors should reconcile these contradictory observations.

We appreciate the Reviewer's helpful comments. The manuscript has been edited in accordance with the Reviewer's recommendations as described below.

The results from our in vitro studies do not indicate that a particular membrane topology *per se* is required for the MakA induced tube formation.

On the contrary, we interpret our data to indicate that MakA can interact with membrane lipids representing either side or topology of a membrane bilayer and thereby induce tubulation when the pH is acidic.

The CD spectrometry results and TEM negative staining images indicate that under acidic conditions, MakA undergoes conformational change, presumably adopting a structure that promotes membrane interaction and the formation of oligomeric assemblies in the presence of membrane lipids. (Figure 3A and Figure 3—figure supplement 1B).

Similarly, we infer that a pH induced conformational change and oligomerization of MakA in lysosomes may lead to lysosomal tubulation (Figure 1A). Western blot analysis of lysosomes isolated from MakA treated HCT8 and Caco-2 cells detected formation of dimeric, tetrameric and oligomeric MakA complexes in addition to the monomer (Figure 1—figure supplement 1B), which is consistent with the notion that MakA oligomerization inside lysosomes may be responsible for lysosomal tubulation.

– Only when pH was acidic in the cell culture medium did the authors detect plasma membrane tubulation in cells, similar to the one they find in model membranes in terms of protein/membrane topology.

As mentioned above, we interpret our data to indicate that MakA can interact with membrane lipids representing either side or topology of a membrane bilayer and thereby induce tubulation when the pH is acidic. The results from our in vitro studies do not indicate that a particular membrane topology *per se* is required for the MakA induced tube formation under acidic pH conditions.

The authors should clarify whether these experimental conditions for the addition of MakA to the extracellular medium at acidic pH would have any physiological relevance in the context of the actual host/pathogen interaction.

With the use of *C. elegans*, which served as a predatory model organism, we proved that MakA is a fitness factor for *Vibrio cholerae* in our previous research (PMID: 30271941). It is well known that the pH of the intestinal lumen of a live *C. elegans* ranges from 5.96 in the pharynx to 3.59 in the lower intestine (PMID: 23668893).As an additional point of interest, we recently discovered that MakA accumulates in the acidic endolysosomal compartment and causes lysosomal dysfunction (PMID: 33720402). Based on these observations, we hypothesized that pH may regulate the activity of MakA. We exposed the target cells to MakA under different pH condition. Consistent with our hypothesis, we observed dramatic increase in the activity of MakA when the pH of the extracellular media was acidic (Figure 1 and 2).

The authors should also address whether membrane tubulation of the plasma membrane induced by MakA induce cell death of the HCT8 cells.

The MakA induced plasma membrane tubulation by MakA under acidic condition results in the loss of plasma membrane integrity and epithelial cell death and erythrocytes hemolysis. MakA induced pH-dependent cell toxicity in a number of epithelial cell lines including HCT8, HCT116 and CaCO2 (Figure 2B and Figure 2—figure supplement 1A,B). Notably, MakA induced plasma membrane tubulation of erythrocytes resulted in lysis of erythrocytes (Figure 2C).

– The authors should explore in more detail the lipid dependency of the membrane alterations induced by MakA, not just using pure PC as a control. They could incorporate for example negatively charged lipids, PIPs, cholesterol and sphingomyelin using the Biacore assay.

As mentioned in a response to Reviewer #2, we decided to remove the preliminary data generated from the Biacore assay. We agree with Reviewer #2 that the results reported in Figure 3A are challenging to interpret for the very reasons given. As it is questionable if these Surface plasmon resonance data can add anything to the story, we decided to omit the experiment and we have removed this figure.

– The authors should clarify if lipids form a continuous bilayer along the tube or are they laterally separated by protein transmembrane domains. If the membrane is not continuous, the authors should address whether the permeability is affected, which would be important in the context of the tubulation observed in cells.

The transmembrane domains of MakA laterally separate the lipid bilayer, and together they form a tightly packed protein-lipid spring. The lipids are not well resolved. There are, nevertheless, distinct lipid densities with unclear identities present. In low-pass filtered volumes, the lipid layer becomes more visible, indicating a continuous protein-lipid tube. A new Figure 4—figure supplement 4, prepared in response to Reviewer #2, also visualizes the innermost and potentially closed-off lipid/protein tube pore. Additional experiments will be required to answer if the inner tubular space is completely closed off from the extra-filamentous space.